# Recent Advances in Lipid Nanoparticles for Delivery of mRNA

**DOI:** 10.3390/pharmaceutics14122682

**Published:** 2022-12-01

**Authors:** Lei Yang, Liming Gong, Ping Wang, Xinghui Zhao, Feng Zhao, Zhijie Zhang, Yunfei Li, Wei Huang

**Affiliations:** 1College of Chinese Materia Medica, Tianjin University of Traditional Chinese Medicine, Tianjin 301617, China; 2State Key Laboratory of Bioactive Substance and Function of Natural Medicines, Department of Pharmaceutics, Institute of Materia Medica, Chinese Academy of Medical Sciences and Peking Union Medical College, Beijing 100050, China; 3Beijing Bio-Bank Co., Ltd., Beijing 100107, China

**Keywords:** messenger RNA, non-viral vectors, viral vectors, lipid nanoparticles

## Abstract

Messenger RNA (mRNA), which is composed of ribonucleotides that carry genetic information and direct protein synthesis, is transcribed from a strand of DNA as a template. On this basis, mRNA technology can take advantage of the body’s own translation system to express proteins with multiple functions for the treatment of various diseases. Due to the advancement of mRNA synthesis and purification, modification and sequence optimization technologies, and the emerging lipid nanomaterials and other delivery systems, mRNA therapeutic regimens are becoming clinically feasible and exhibit significant reliability in mRNA stability, translation efficiency, and controlled immunogenicity. Lipid nanoparticles (LNPs), currently the leading non-viral delivery vehicles, have made many exciting advances in clinical translation as part of the COVID-19 vaccines and therefore have the potential to accelerate the clinical translation of gene drugs. Additionally, due to their small size, biocompatibility, and biodegradability, LNPs can effectively deliver nucleic acids into cells, which is particularly important for the current mRNA regimens. Therefore, the cutting-edge LNP@mRNA regimens hold great promise for cancer vaccines, infectious disease prevention, protein replacement therapy, gene editing, and rare disease treatment. To shed more lights on LNP@mRNA, this paper mainly discusses the rational of choosing LNPs as the non-viral vectors to deliver mRNA, the general rules for mRNA optimization and LNP preparation, and the various parameters affecting the delivery efficiency of LNP@mRNA, and finally summarizes the current research status as well as the current challenges. The latest research progress of LNPs in the treatment of other diseases such as oncological, cardiovascular, and infectious diseases is also given. Finally, the future applications and perspectives for LNP@mRNA are generally introduced.

## 1. Introduction

Since the outbreak of coronavirus disease 2019 (COVID-19) in early 2020, severe acute respiratory syndrome coronavirus 2 (SARS-CoV-2) has been spreading at an alarming rate worldwide, causing deaths from acute respiratory distress syndrome and multiple organ dysfunction syndrome [1] and seriously endangering the health of people worldwide. With over 250 million confirmed cases of COVID-19 and over 5.08 million deaths by 2022, the initial attempt to develop conventional vaccines was difficult due to the limitations of the adult targets and the long development cycle.

In this increasingly serious epidemic, mRNA technology holds the promise and has changed the landscape of conventional medicine in just one year with the young but powerful triumvirate of mRNA companies now in place: Moderna Therapeutics, which was founded in 2010 and marked the largest biotech IPO in history at the end of 2018, spent only 45 days from identifying the viral antigen sequence to producing the first mRNA vaccine for clinical use with its new vaccine using mRNA technology [2]; Pfizer/BioNTech, Europe’s largest biotech unicorn, which closed the largest funding round ever raised by a European biotech company on 10 July 2019, produced the world’s first approved new coronavirus vaccine with its mRNA vaccine Comirnaty (BNT162b2, an LNP-formulated, nucleoside-modified RNA vaccine that encodes a prefusion stabilized, membrane-anchored SARS-CoV-2 full-length spike protein); and CureVac, which was founded in 2000, is the leading company in the mRNA pharmaceutical industry, and has its own patent on ionizable lipids. These three companies are far ahead in the field of mRNA technology.

In December 2020, Moderna and BioNTech’s mRNA vaccines for COVID-19 (BNT162b2 and mRNA-1273 (an LNP-encapsulated mRNA-based vaccine that encodes for a full-length, prefusion-stabilized spike (S) protein of SARS-CoV-2)) received emergency approval from the Food and Drug Administration (FDA) due to the excellent phase III clinical trial results. The effectiveness of these mRNA vaccines against COVID-19 was 95% and 94.1%, respectively, demonstrating the high effectiveness of the mRNA vaccine against the novel coronavirus pneumonia [3,4]. The significant progress made by these three mRNA giants in producing an mRNA vaccine for COVID-19 has sparked a global surge in mRNA therapeutic research; mRNA technology was ranked at the top of the *MIT Technology Review*’s list of the world’s “Top 10 Breakthrough Technologies” in 2021. In the midst of this sudden and complex outbreak, there was a growing consensus that an mRNA vaccine might be the best weapon against the virus’ mutations [5].

There were 33 COVID-19 vaccines approved worldwide as of 24 January 2022 for use in 197 countries, of which 10 of the vaccines were approved by the World Health Organization (WHO) for emergency use. In addition, as of 25 January 2022, a total of 194 vaccines were in preclinical development and 140 were in clinical development worldwide [6], Among these, the mRNA-based vaccines are shown in Table 1.

Why did the mRNA vaccine emerge among the various new COVID-19 vaccines? Zeli Zhang et al. [7] analyzed four COVID-19 vaccines from three different vaccine platforms; namely, an mRNA platform (Pfizer/BioNTech BNT162b2 vaccine and Moderna mRNA-1273 vaccine), a recombinant protein-based adjuvant vaccine platform (Novavax NVX-CoV2373 vaccine SARS-CoV-2 spike), and a viral vector-based platform (Janssen/Johnson & Johnson Ad26.COV2.S vaccine) for immune responses triggered by the same pathogen followed by a 6-month longitudinal examination. In terms of antibodies, participants who received the Moderna vaccine had the highest levels of neutralizing antibodies, followed by those who received the Pfizer/BioNTech and Novavax vaccines, while the Janssen/Johnson & Johnson vaccine had the lowest levels of neutralizing antibodies. In addition, all vaccinees retained memory CD4^+^ T cells to fight the virus, but participants who received the Moderna and Novavax vaccines carried higher levels of follicular helper T cells and cytotoxic CD4^+^ T lymphocytes, and those who received the Pfizer/BioNTech, Moderna, or Janssen/Johnson & Johnson vaccines showed higher CD8^+^ T responses. Overall, only 60–70% of the participants retained the memory CD8^+^ T cells after six months. This study disclosed that most people maintained certain immune responses to SARS-CoV-2 regardless of the applied vaccine types and that among them, the mRNA vaccines were undoubtedly the most effective.

The reasons that mRNA vaccines are so powerful lies in the many advantages of mRNA therapy technology: (1) Safety. In contrast to DNA, which needs to enter the nucleus, mRNA functions in the cytoplasm, thereby avoiding the risk of its integration into the host genome [8]. (2) High efficiency. Appropriate modification regulation and sequence optimization can provide a significant increase in mRNA stability and translation efficiency [9,10,11], and an efficient delivery system has been developed to enable rapid uptake and cytoplasmic expression of mRNA. (3) Short development period. Once the genome sequence of the pathogen is determined, the mRNA that encodes the antigenic protein can be designed [12], and the transcript of the mRNA is produced in vitro without the need for cellular amplification, which significantly increases the speed of production. (4) High flexibility. In case of mutation of the virus whereby the original vaccine becomes ineffective, a new mRNA vaccine can be rapidly redesigned and produced based on the new viral sequence. (5) Wide range of applications. mRNA technology is making breakthroughs in the treatment of infectious diseases, genetic diseases, cancer, diabetes, etc., by virtue of its advanced principles and high efficiency in research and development and production.

This review focuses on the progress of mRNA delivery systems. We present the disadvantages of viral vector delivery systems and summarize the advantages of non-viral vectors, especially lipid nanoparticles (LNPs), the status of research, and the future insights for optimization. Finally, our thoughts on future LNP@mRNA delivery systems are shared.

## 2. The Vectors for mRNA Delivery

### 2.1. The Delivery of mRNA Requires Vectors

The recent success of BNT162b2 (COMIRNATY vaccine from BioNTech/Pfizer) and mRNA-1273 (Spikevax COVID-19 vaccine from Moderna) against the new COVID-19 outbreak was a result of the meticulous planning and coordinated efforts of all the teams involved in the development process. However, this could only be achieved due to the solid foundation established by decades of pioneering work by those who saw mRNA as a therapeutic drug. Since Francois Jacob et al. first reported mRNA in 1961 [13], mRNA has been recognized as a bridge between the DNA coding sequence and the production of functional proteins. Subsequently, in 1990, Wolff J. et al. laid the foundation of mRNA as a therapeutic agent by injecting naked mRNA containing reporter genes into the skeletal muscle of mice and providing a proof of principle for direct gene transfer in vivo [14]. In the following decades, there have been significant advances in the understanding of mRNA and how it can be used as a therapeutic agent [15].

However, although animal studies have shown that mRNA can act like a vaccine for the treatment of infectious diseases and cancer, mRNA’s unstable nature, short half-life, and susceptibility to enzymatic degradation make it difficult to realize its therapeutic potential [16,17]. However, due to the persistence of several mRNA experts, the concept has been turned into a reality. The fundamental research pioneered by Katalin Kariko, her colleague Drew Weissman, and many others in the mRNA field has demonstrated the importance of UTR selection, codon optimization, nucleoside modification, purification, end-capping techniques, and length of poly A tails in the stability and translatability of mRNA as well as in reducing immune stimulation.

In addition, as a nucleic-acid-like biologically active macromolecule with a molecular weight of 20 to several thousand bp, mRNA has more negative charges and is more difficult to encapsulate and release from endosomes but is highly degradable in vivo. Although naked mRNA can still be taken up by cells, this process is inefficient and the use of commercially available transfection reagents for in vivo delivery of mRNA has had limited success [15]. The delivery of mRNA via mechanical methods such as gene gun delivery and electroporation is theoretically possible, but their application is limited because they tend to induce tissue damage [18].

At the same time, there are three barriers that need to be breached for mRNA to enter and successfully express in target cells: (1) Extracellular barrier. mRNA is easily degraded by the RNA enzyme (RNase) in the body and may also be recognized and phagocytosed by macrophages or dendritic cells in the liver, making it difficult to transport to other tissues and organs. (2) Lysosomal escape. Carried by carriers, mRNA enters the cell via cytosol and needs to be released from the lysosome before it can bind to the ribosome. Meanwhile, there are also receptors in the endosome that recognize mRNA and will send it for degradation. (3) Intracellular immunity. Cells associated with innate immunity (macrophages) have intracellular Toll-like receptors (TLRs) or RLR receptors (RIG-I-like receptors) that can recognize foreign mRNA and then initiate the expression of inflammatory factors such as interferon and clear foreign mRNA [19].

Therefore, as of the 1990s, the development of suitable mRNA delivery systems was the key to the clinical application of mRNA therapeutics and was the first problem to be solved [20]. Currently, there are two main types of mainstream delivery vectors: viral and non-viral vectors. Transfection efficiency is one of the key factors in the development of nucleic-acid-based drugs, so viral vectors are extremely attractive due to their unique transfection efficiency. Due to the toxicity of viral vectors and the rapid development of materials science and biomedical technology in recent years, an increasing number of gene therapy drugs with non-viral vector materials are entering clinical trials, especially organic materials, which stand out in the arena of gene therapy due to their special properties and advantages. However, the low escape efficiency of drugs delivered via non-viral vectors in endosomes or lysosomes [21,22] and their weak targeting in cells, tissues, and organs are challenges that need to be urgently overcome in the current non-viral vector delivery systems. The advantages and disadvantages of the current viral and non-viral vectors are shown in Table 2.

### 2.2. Selection of mRNA Delivery Vectors

#### 2.2.1. Viral Vectors

Viruses are the smallest and simplest living parasites without a cellular structure. Due to their ability to efficiently infect human cells and their molecular mechanism for delivering their genome into the cell, viruses are far more utilized as delivery vectors than non-viral vectors, with approximately 70% of gene therapy using viruses as delivery vectors. However, most viruses are pathogenic and must be artificially modified to retain only their own DNA-integrated functional elements while eliminating the original pathogenic functional elements. Although most of the currently marketed gene therapy drugs use viruses as vectors for gene delivery, among which the most common viral vectors are retroviruses, adenoviruses, adeno-associated viruses, and lentiviruses [23], viral vectors have disadvantages such as high immunogenicity, large safety risks, and difficulties in production [24].

However, the boom in gene therapy using viral vectors came to an abrupt end when Jesse Gelsinger, an 18-year-old patient with ornithine transcarbamyltransferase deficiency (OTC), died four days later due to multiple organ failure in a phase I clinical trial led by James Wilson of the University of Pennsylvania after researchers injected approximately 1 trillion adenoviruses carrying the gene encoding the therapeutic OTC directly into his liver, which subsequently caused a strong immune rejection reaction, resulting in the first death directly related to gene therapy [25]. One year later, 17 children with SCID received gene therapy with retroviral vectors in France, two of whom developed secondary acute leukemia [26]. Since then, the clinical development of gene therapy has hit rock bottom, but it has also forced attention to the development of non-viral gene vectors.

#### 2.2.2. Non-Viral Vectors

Non-viral vectors are compounds that are positively charged under physiological conditions and include cationic polymorphs, cationic liposomes, etc. They interact with negatively loaded mRNA and enter the cell via endocytosis through the principle of charge interaction. Without the disadvantages of virus-based vectors, they are easy to prepare and are not immunogenic, but they are much less efficient than virus-based vectors when introducing exogenous genes. A good non-viral vector delivery should have the following characteristics: (1) prevention of nuclease-mediated degradation, (2) maximization of cell-specific uptake, (3) efficient endosomal escape, (4) effective clearance of delivered vector components, and (5) maximization of delivery efficiency and minimization of immunological effects.

In order to combine the advantages of the high transfection efficiency of viral vectors with the safety of non-viral vectors, Yujia Cai et al. [27] invented a delivery technique for a virus-like particle (VLP) between viral and non-viral vectors using the principle of specific recognition of the mRNA stem-loop structure and phage capsid protein through virus engineering technology, which could deliver CRISPR/Cas9 mRNA. It was found that VLP-mRNA could significantly reduce—and even completely avoid—off-target effects compared to viral systems that expressed Cas9 for extended periods of time. In addition, VLP-mRNA could deliver the entire CRISPR component (Cas9 with gRNA), thereby overcoming the limitations of the adeno-associated virus (AAV) vector’s small carrying capacity and even delivering larger base editing tools. Using the delivery platform, the team conducted a preclinical study of CRISPR in vivo gene editing for the treatment of viral keratitis and effectively inhibited HSV-1 virus replication in successfully curing a model mouse of herpetic keratitis and preventing viral recurrence [28]. However, given that this technology is still in its early stages, its safety and effectiveness have yet to be proven in more studies.

Currently, non-viral delivery systems for mRNA have evolved into the third generation; a comparison of the different delivery systems is shown in Table 3. The first-generation RNA delivery systems used delivery vehicles that appeared in the 1990s such as protamine [29,30], polyethylenimine (PEI), and cationic liposomes [31]. The second generation of delivery systems were mainly the biodegradable, ionizable polymers that emerged in the late 1990s [32,33]. Most of these materials failed to enter clinical research due to their own high toxicity, complex structure, and uncontrollable polymerization. For example, early Moderna studies mainly used protamine to deliver mRNA, but the mRNA was too tightly bound to the arginine in the protamine; one study showed that the size of the free mRNA in dynamic light scattering experiments was close to 50 nm, while the protamine–mRNA complex was in the range of 250–350 nm [34], which reduced the protein expression and was detrimental to the release of mRNA [35,36]. Although this approach was adopted for CureVac’s rabies vaccine candidate CV7201, in a phase I human trial in which 80–640 μg doses were administered by the subcutaneous and intramuscular routes, one of 101 participants suffered facial muscle weakness or paralysis (Bell’s palsy) at the high dose, while 5% of the participants also suffered from severe adverse reactions. There was a high overall incidence of adverse reactions, 97% of which occurred at the injection site and 78% of which occurred systemically. Owing to the unsatisfactory therapeutic effect of mRNA delivery via protamine and its serious adverse effects, CureVac eventually abandoned this method [37]. Cationic polymers such as PEI are rarely used for mRNA delivery due to the complexity of the molecular structure and the uncontrollable degree of polymerization [38,39,40]. Haifa Shen from Stemirna Therapeutics developed a lipopolyplex nano-delivery platform (LPP) with polymer-encapsulated mRNA as the core and a phospholipid-encapsulated bilayer structure. They claimed that the bilayer nanoparticles of the LPP provided better mRNA encapsulation and progressive release of mRNA molecules with polymer degradation as well as excellent dendritic cell targeting for better activation of T-cell immune responses through antigen presentation [41]. Using the SW-BIC-213 delivery platform, a new COVID-19 vaccine has been developed and is currently in phase I trials [42]. Furthermore, to enhance the stability of mRNA, Jacob A. Poliskey et al. utilized a PEGylated polyacridine peptide to bind double-stranded mRNA (dsmRNA) to generate dsmRNA polyplexes that were metabolically stable in the circulation. In addition, Yoshinaga et al. complexed mRNA with derived poly(ethylene glycol)-poly(cationic) block copolymers with phenylboronic acid and polyol groups to obtain polycomposite micelles (PM) with ATP-responsive cross-linked mRNA loading in the inner core to prevent enzymatic degradation [43,44].

Compared to conventional cationic polymeric carriers, lipid nanoparticles (LNPs) made with ionizable lipids have shown many advantages: good biocompatibility, high nucleic acid encapsulation and effective transfection, high tissue penetration for delivery of therapeutic agents, intelligent drug release, low off-target effects, and low cytotoxicity and immunogenicity. Hence, LNPs have become the most studied and widely used third-generation mRNA delivery system [45,46], which has greatly contributed to the development of the mRNA industry and clinical applications such as the application of LNP-assisted mRNA delivery for potent cancer immunotherapy [47]. As shown in Table 3, for nucleic acid drugs, LNPs have more advantages than other delivery systems in terms of the encapsulation effect, expression efficiency, and safety in vitro and vivo. It can be said that lipid nanoparticles (LNPs) are the best delivery system at present.

**Table 3 pharmaceutics-14-02682-t003:** Advantages and disadvantages of various vectors.

	Advantages	Disadvantages
Retrovirus	Stable integration into the genome, high transfection rate, long gene expression time, weak immunogenicity.	Only integrate into dividing cells, risk of insertional mutations, low delivery efficiency in vivo, small package capacity [9].
Adenovirus	Capable of carrying relatively large gene fragments.	Immunogenic, complex operation, short gene expression time.
AAV	Weak immune response, high transfection rate, no integration of host DNA.	Small package capacity
Lentiviral	Stable integration into genome or dividing cells, long duration of gene expression, weak immunogenicity.	Risk of insertional mutations, low delivery efficiency in vivo [9].
LNP	Protect mRNA from degradation by ribonucleases, high mRNA delivery efficiency, high yield, and easy scale-up of production.	Potential side effects, weak targeting and stability.
Protamine	Protect mRNA from degradation by ribonucleases and adjuvant activity of the protamine–mRNA complex.	Low delivery efficiency and low efficiency of mRNA translation.
Cationic polymer	Promote internalization through adsorption-mediated cellular endocytosis, effectively compress nucleic acids and protect them from enzymatic degradation through surface amine groups [48], and provide pH-buffering capacity through the large number of tertiary amine groups in the core [49].	Mostly non-degradable and highly cytotoxic [50].
Peptide	Highly functional.	Low delivery efficiency
Cationic nanoemulsion	Protects mRNA from degradation by ribonucleases, has the ability to protect and efficiently deliver nucleic acids, and can trigger a strong immune response as a vaccine adjuvant [51].	High cytotoxicity
Cell-penetrating peptide	Low charge density and excellent ability to cross cell membranes.	Only a few peptides are effective and there is an urgent need to develop new effective compounds to expand the material pool for peptide delivery systems.
Exosome	Biocompatible and not easily cleared by immunity [52].	Difficult to produce, isolate, and purify [53].
Inorganic nanoparticle	Easily modified for surface modification and unique versatility.	Poor biocompatibility and difficult to biodegrade.

## 3. Lipid Nanoparticles (LNPs)

### 3.1. The Development of LNPs

LNPs were originally adopted for the delivery of siRNA because siRNA only needs to be delivered into the cytoplasmic matrix, where it is recognized and bound by the RNA-induced silencing complex (RISC) to induce silencing of the target mRNA. In 2006, Tracy S. Zimmermann et al. [54] first successfully demonstrated that the application of LNPs for the systemic delivery of siRNA could downregulate gene expression in non-human primates, thereby knocking down serum apolipoprotein B levels. The success of this clinical trial indicated the feasibility of the LNP delivery of siRNA in macaques and the potential of LNPs as a nucleic acid delivery vector in higher-species animal models for selective regulation of gene expression. Then, the development of siRNA-LNP drugs went into high gear until the approval of Alnylam’s siRNA-LNP drug patisiran (ONPATTRO™) in 2018 became a key milestone for siRNA and LNP technology; the drug was used to target the liver for the treatment of hereditary thyrotropin-mediated amyloidosis [55]. It was the successful application of LNPs to siRNA that led to the boom in LNP technology.

The initial success of siRNA-LNP therapeutics [39,56] resulted in interest in the potential application of LNP delivery of other nucleic acid drugs, especially mRNA, and provided valuable experience in the current development of mRNA drugs [57]. Pardi et al. [58] showed that the mRNA encoding the reporter gene firefly luciferase was successfully expressed in liver and other tissues via various routes of administration and expressed persistently (up to 10 days) by intramuscular and intradermal administration; now the collaboration between CureVac and Acuitas has first proved that similar LNPs are active in large animals (pigs and macaques) [59], which provided a good basis for clinical application in humans. Currently, LNPs have developed into a mature technology platform for the delivery of nucleic acid drugs, vaccines, or gene-editing tools. There are also huge potential applications for LNPs in cancer vaccines, protein-replacement therapies, and gene-editing components for rare genetic diseases.

### 3.2. Components and Structural Features of LNPs

An LNP is a spherical vesicle composed of ionizable lipids that are positively charged (bound to RNA) at low pH and neutral at physiological pH with reduced toxic effects compared to continuously positively charged lipids (liposomes). As shown in Figure 1, the LNP@mRNA platform is a multi-component system that consists of ionizable lipids (which bind to negatively charged mRNA and facilitate its endosomal escape and transfection in vivo), neutral phospholipids (co-lipids) (which form a monomolecular phospholipid layer of LNP to facilitate cell binding and destabilize endosomes and improve nucleic acid delivery efficiency), cholesterol (which fills gaps between lipids, stabilizes LNP structure, regulates the fluidity of membranes, and improves particle stability), and PEGylated lipids (which reduce the clearance of serum proteins and the reticuloendothelial system (RES), reduce LNP particle aggregation or fusion, and enhance LNP spatial stability) [60]. In addition, there are stabilizers such as sucrose or alginate that improve the stability of LNP and mRNA vaccines and prevent excessive lipid viscosity.

The proportion of components can be varied according to the target tissue, and the physical properties of LNPs such as the particle size, morphology, encapsulation efficiency, and surface charge can also be adjusted by adjusting the lipid composition [62,63]. For example, the molar ratio of lipid in the organic phase of LNPs affects the size, polydispersity, and efficacy of LNPs. The ratio of lipid to mRNA dosing affects the encapsulation efficiency; most LNPs are formulated with a lipid-to-oligonucleotide weight ratio of 10:1. The molar ratio of nitrogen in ionizable lipids to phosphate in mRNA (N:P) indicates the charge balance between the cationic tertiary amine of the ionizable cationic lipid and the anionic phosphate group of the mRNA backbone. This property is the basis for complexation of ionized cationic lipids with mRNA; the N:P ratio of an LNP is usually around 6. The lipid acid dissociation constant (p*K*a) is the pH of the ionized and non-ionized forms of lipids at the same concentration, which affects the encapsulation efficiency, efficacy, delivery, and toxicity of LNPs. For mRNA delivery, the lipid p*K*a is generally between 6 and 7. Additionally, the optimal lipid p*K*a ranges for intravenous and intramuscular administration are 6.2–6.6 and 6.6–6.9, respectively, in different routes of administration.

Three important parameters of the LNP aqueous phase buffer are composition, ionic strength, and pH. The buffer can stabilize mRNA in solution; ionizable lipids, when mixed in the acidic buffer, undergo protonation and generate a positive charge, and in turn complex with mRNA. A common buffer used in LNP preparations is 25–50 mM sodium acetate or sodium citrate at pH 4–5 with final storage and use in phosphate-buffered saline (PBS) at pH 7.4.

The typical structure of LNP@mRNAs is shown in Figure 2, including a nanostructured core with a homogeneous core shell. The morphology of LNP cores can be classified as multi-layered vesicle structures or other self-assembled morphologies, but the fine structure of the complexes formed by the lipids and mRNAs inside the particles is unclear; more studies are needed to confirm the morphology [64,65,66].

### 3.3. Preparation of LNPs

Lipid nanoparticles can be classified according to their structure and drug-delivery mechanism: solid lipid nanoparticles, nanostructured lipid carriers, lipid nanocapsules, lipid–drug conjugates, polymeric lipid-hybrid nanoparticle lipids, etc. [67]. Different types of LNPs have different preparation methods such as thin film hydration, solvent injection, ultrasound-assisted, high-shear homogenization, solvent evaporation, solvent diffusion, etc. [68]. The preparation method determines the properties of the LNPs, including their size, homogeneity, and encapsulation efficiency. The choice of preparation method should be accompanied by considerations of cost, scalability, reproducibility, and time. The current mainstream laboratory methods for the preparation of LNPs are shown in Table 4.

#### 3.3.1. Ethanol Dilution Method and Manual Mixing Method

Ethanol dilution is the simplest of the methods currently available for the preparation of LNPs. Various lipids are dissolved in organic solvents (ethanol, acetone, or isopropanol), and the mRNA is dissolved in an appropriate buffer solution such as an acetate, citrate, or malate buffer; then the two phases are rapidly mixed. As the ethanol phase is diluted, the lipid solubility decreases and gradually precipitates out of the mixed solution to coagulate and form lipid nanoparticles with efficient encapsulation of mRNA. Finally, the residual ethanol is removed via dialysis or ultrafiltration and the pH of the buffer is adjusted [69].

Manual mixing is a simpler alternative to the ethanol dilution method by transferring the ethanol lipid mixture into an aqueous acidic mRNA solution, mixing quickly by pipetting for 15 s, and finally leaving the mixture to stand for 10 min. As with the ethanol dilution method, manual mixing results in a non-homogeneous LNP with low encapsulation efficiency and problems of poor reproducibility, variable results, and poor particle size uniformity.

#### 3.3.2. T-Mix Method

To avoid the shortcomings of the ethanol dilution method, some studies further optimized the method by mixing the solution with an aqueous solution containing nucleic acids at an acidic pH using a T-mixer. When these two solutions are mixed, the total solubility of the lipids is decreased via the dilution with ethanol, and the ionized lipids become positively charged, bind to the negatively charged nucleic acids, and self-assemble to form complexes. The advent of the T-mixer provided a controlled mixing situation for the preparation of LNPs [70] with carefully designed microscopic features within the channels that can mix the two fluids together in a controlled and reproducible manner.

#### 3.3.3. Microfluidics

Microfluidics is the most suitable technique for the preparation of LNPs. The method, which is relatively simple and rapid, has mild conditions, and easily achieves scale-up in production, currently is becoming one of the most common methods for the preparation of LNPs at a small scale in the laboratory. Compared to conventional methods, microfluidic methods can produce homogeneously sized LNPs with high reproducibility. Furthermore, LNP production conditions optimized at a laboratory scale were unchanged when used for large-scale production. These findings suggested that microfluidic devices accelerated the development of LNP-based nanomedicines.

The principle of microfluidics is as follows (Figure 3): after dissolving lipids and nucleic acids in the organic and aqueous phases, respectively, the two-phase solution is injected into the two inlet channels of the preparation system with an aqueous solution of mRNA at one end and an ethanol solution of lipids at the other; the synthesis of nucleic acid lipid nanoparticles is completed via rapid mixing of the two phases. The two fluids mix completely within a millisecond, resulting in a change in the solvent polarity and thus triggering the self-assembly of nucleic-acid-laden lipid nanoparticles. The technique, which is suitable for most LNP formulation and the payload of RNA, can achieve high load-encapsulation efficiency (>80%) [71] with increased reproducibility, stability, and scalability of the LNP structure. Additionally, the appearance of various microfluidic chips such as those with fluid channels with specific geometries (sinuous shapes) that trigger local chaotic turbulence to accelerate mixing has made it easier to prepare LNPs. In addition, microfluidic chips also have zigzags, vortex configurations, branching structures, etc.

Microfluidics can control the particle size of lipid nanoparticles by changing the rate and ratio of fluid injection. When ethanol is rapidly diluted with a buffer to a critical ethanol concentration, small-sized LNPs are formed; while large-sized LNPs are formed under slow ethanol dilution conditions [72]. Currently, most of the studies in the literature used microfluidics to prepare LNPs, and the particle size was generally distributed between 20 and 200 nm. Nanoparticles with a particle size of less than 20 nm are easily metabolized by the kidney, a large particle size has a greater impact on the transfection effect and safety. As influenced by the particle size, LNPs can only pass through porous endothelial cells, making them the best choice for targeting the liver [31,32,33]. Belliveau et al. achieved reproducible production of LNP-siRNA systems with a 20 nm diameter or larger with a polydispersity index as low as 0.02 using microfluidic mixing at the nanoliter scale with millisecond mixing [34]. LNPs with diameters of 70 nm or larger can also be prepared via microfluidic mixing techniques that use a macroscopic mixing process. Therefore, microfluidic mixing is a very precisely controlled and easily scalable technology that will become the primary method for formulating LNP delivery systems.

It also was reported that large-scale production could be achieved by using an impinging-jet mixing method similar to microfluidics. A BlueShadow 80P high-pressure pump was employed to allow the vaccine and lipid solution to form two jets that hedged in the cavity, thereby allowing the individual lipid components to mix sufficiently via the fluid dynamics to form liposome nanoparticles that encapsulated the mRNA. The delivery of mRNA via microfluidic devices is becoming a key and core technology for the next generation of RNA, cell, and gene therapies.

## 4. The Factors Affecting the Efficiency of LNP@mRNA Delivery

In addition to the preparation of the LNPs, the modification of the mRNA and the relative amounts of ionizable lipids, auxiliary lipids, cholesterol, and PEG also influence the efficacy of the lipid nanoparticles. The optimization of the given application and route of administration is also required. In addition, the category, size, and surface charge of lipids also affect the behavior of lipid nanoparticles in vivo, so the optimization of lipid nanoparticle formulations for nucleic acid delivery has a long way to go.

### 4.1. Modification of mRNA

In the last decade, new technological developments have gradually arisen to enhance mRNA stability and translation efficiency (Figure 4) as well as to modulate the immunogenicity of mRNA vaccines while reducing immune stimulation, mainly by exploiting the importance of UTR selection, codon optimization, nucleoside modification, purification, end-capping techniques, and length of the polyadenosine (poly-A) tail in mRNA stability and translatability.

Nowadays, a variety of modification techniques are available to produce more stable mRNA: these can be broadly classified as: replacing natural RNAs with synthetic unnatural RNAs to synthesize mRNA, adjusting the N^7^-methylguanosine (m^7^G) linked to the first nucleotide of the mRNA via a reverse 5′ to 5′ triphosphate bond (Cap) (m^7^GpppN) (which can enhance mRNA stability and improve protein expression), optimizing the regulatory elements of the 5′ untranslated region (UTR) and 3′ UTR (which enhances the stability of mRNAs and improves their translation efficiency and extends their half-life), codon optimization (replacing the rarer codons in mRNA with more common synonymous codons can improve protein expression), and adding poly-A (which can enhance mRNA stability and translation efficiency).

#### 4.1.1. Nucleoside Modification

The replacement of natural RNA with synthetic unnatural RNA to synthesize mRNA is currently one of the most common approaches. Kariko et al. showed that the incorporation of naturally occurring, chemically modified nucleosides such as pseudouridine (Ψ), thiouridine (s2U), and 5-methylcytidine (m5C) made them more like natural mRNA and significantly reduced the immunogenicity and instability of the mRNA [74], for which the most common method was pseudouridine (Ψ) nucleoside modification to significantly increase the stability and protein expression. In 2018, a study conducted by Mauger et al. using the Zika virus vaccine in mice showed that vaccination with LNP@mRNA that encoded nucleoside modifications (1-methyl pseudouridine, m1Ψ) of various viral surface antigens resulted in stronger protein expression and T- and B-cell responses compared to LNPs that carried unmodified mRNA [75]. Meanwhile, m1Ψ-modified mRNA increased the base build-up and melting point of mRNA, resulting in better mRNA stability. Moderna and BioNTech have also used m1Ψ-modified mRNA to enhance the stability and antigen expression of the new crown vaccine while reducing the inflammatory response to the vaccine [76]. In contrast, CureVac’s sequence-optimized vaccine candidate (CVnCoV) consists entirely of unmodified nucleosides, an approach that enables sequence engineering with unmodified mRNA to provide a robust and balanced immune response. However, the preliminary phase IIb/III data suggested that this vaccine candidate was significantly less effective than the other two leading mRNA vaccines from Pfizer/BioNTech and Moderna with a protection rate of only 47% [77]. Although the poor efficacy of the CureVac vaccine candidates can be attributed primarily to the inclusion of unmodified nucleosides, variation in COVID-19 strains, and the age of the recipients, several other differences between vaccine candidates should be considered such as differences in non-coding elements and storage conditions. Thus, it appears that the selection of an unmodified mRNA to produce the COVID-19 vaccine may not be an appropriate option. Interestingly, in CV2CoV, a second-generation COVID-19 vaccine currently being developed by CureVac in collaboration with GlaxoSmithKline (GSK), although it also uses unmodified mRNA, the results of animal studies in rats and monkeys showed that the vaccine was able to produce approximately 10-fold more neutralizing antibodies after design adjustments, so it is too early to conclude that unmodified natural mRNAs are unsuitable for use in vaccines. Therefore, when considering nucleotide modifications, there is a trade-off between the potential innate adjuvant response driven by unmodified nucleosides and the protein expression caused by modified nucleosides.

To further investigate the effects of these chemical modifications on mRNA, researchers [78] recently proposed a strategy for the dual chemical modification of mRNA by introducing a third base pair during the mRNA transcription, resulting in the insertion of synthetic nucleotides in the 3′ UTR and introducing unnatural nucleotides into specific sites in combination with the natural mRNA of the Ψ and 5 mC (methylation modifications) modification methods. It was found that the dual chemical modifications of the mRNAs could create synergistic effects that improved the mRNA stability, efficiency, and protein expression. This approach may open up new avenues to mRNA modification in which the properties of mRNAs are finely tuned and delivered efficiently into the cell via a covalent vector system. The authors also indicated that by introducing artificial regulatory elements in the untranslated regions of the mRNA, the team may explore the potential to modulate the translation level of mRNAs.

Recently, Dong Yizhou’s team at Ohio State University highlighted efforts in mRNA engineering research areas [79]. Nucleotides were chemically modified to optimize the chemical structure of mRNA to increase the protein expression, including Ψ, 5-methoxyuracil (5 moU), and N1-methyl pseudouracil (me1Ψ). In addition, sequences of 5′ UTR and 3′ UTR of the mRNA were designed to significantly enhance the protein expression. With the development of LNPs and mRNA engineering, these technologies will be applied to the treatment of genetic diseases, infectious diseases, cancer, and other diseases. For example, TT3 and its analogue-derived lipid-like nanoparticles could effectively deliver coagulation factor IX or VIII-mRNA and restore coagulation activity in hemophiliac mice. The mRNA that encodes the SARS-CoV-2 antigen could be a candidate vaccine for COVID-19. Vitamin lipid nanoparticles containing the antimicrobial peptide histone and cathepsin B (AMP-CatB) mRNA could enhance the transfer of macrophages to treat multi-drug-resistant bacterial sepsis [80]. Mimetic lipids such as phospholipids that encode mRNAs for co-stimulatory receptors may enhance cancer immunotherapy. In a recent work, Yin Yu et al. [81] induced regeneration of cartilage tissue without blood vessels via in situ injection of modified mRNA that encoded growth factors and enhanced the efficacy of bone marrow mesenchymal stem cells in healing cartilage injury, proving that modifying mRNA to enhance stem cell therapy is a very efficient and safe therapeutic option with a potential clinical translation.

#### 4.1.2. Adjusting the Cap Structure

The mRNA addition cap significantly improves the translation efficiency and the intracellular mRNA stability by binding to eukaryotic translation initiation factor 4E. The capping of mRNA is usually carried out with a cap analogue, which can be added during or after IVT transcription. However, mRNA can be reverse-capped, leading to rapid degradation and poor translation. To avoid reverse 5′ cap adulteration, an anti-reverse cap analogue (7-methyl (3′-O-methyl) (5′) Gppp (5′) guanosine; m7G (5′) ppp (5′) G) (ARCA) has been developed that ensures a correct capping orientation. Further improvements have been made over the years in order to improve the performance of ARCAs. Therefore, for an effective LNP@mRNA vaccine, the incorporation of a stable and correctly oriented cap is required.

For the rapid and large-scale production of mRNA, TriLink recently developed CleanCap™, a novel co-transcriptional capping method starting with 5′ AU. This method produces a natural (7m)GpppN(2′-Om) (cap-1) structure that improves the translation of mRNA to functional proteins compared to (7m)GpppN (cap-0) obtained via traditional capping methods (the anti-reverse cap analogue ARCA), which has been shown to activate the pattern-recognition receptors involved in the breakdown of non-self RNAs [82,83,84]. In addition, CleanCap™ technology eliminates the cost and sample loss associated with traditional enzyme-capping methods such as ARCA and mCap. In addition, Blue Magpie developed its own third-generation Cap1 chemical capping technology that allows for the rapid and stable synthesis of mature mRNA carrying a 5′-Cap1 structure with an mRNA purity of more than 99%.

#### 4.1.3. Optimization of Regulation Elements for 5′ UTR and 3′ UTR

Modification and selection of the 5′ and 3′ UTRs that flank the coding sequence have been shown to greatly affect the stability and translation efficiency of the exogenous mRNA. Features of the 5′ UTR such as the start codon and secondary structure may affect ribosome recruitment, scanning, and start codon recognition and thus should be avoided. Overall, the 5′ UTR sequence is essential for protein expression, while the 3′ UTR may affect mRNA half-life. A common approach is to select UTRs from long half-life proteins synthesized from endogenously discovered mRNAs that lack activity such as α and β globin proteins, which are functional proteins that are found in abundance in erythrocytes. The optimized motifs have been proved to produce superior antigen-specific immune responses by increasing the intensity and duration of gene expression in the lymphatic tissues of mice [85,86,87].

#### 4.1.4. Design of the Open Reading Frame (ORF)

Due to the recognition of cellular sensors, the design of the ORF sequence also has an important impact on the translation efficiency and immunogenicity of mRNAs. Apart from the above-mentioned nucleoside modifications, codon optimization has been shown to enhance protein expression by combining frequent codons and codons with higher tRNA abundance to more efficiently utilize cytoplasmic tRNAs [88]. Bypassing codon simplicity and enriching guanine–cytosine (GC) content has now been shown to increase protein levels in vivo [59]. On the basis of this, the CvnCov vaccine adopted unmodified mRNA and only optimized the nucleotide sequence and codon sequence by improving the GC content of the mRNA to achieve an enhanced mRNA stability and immunogenicity [89]. In addition, the optimization of the mRNA sequence is also important because it determines the secondary structure, which can influence mRNA degradation through hydrolysis. Specially designed algorithms have been reported to design optimal mRNA sequences for regions of maximal base stacking, thereby improving mRNA stability.

#### 4.1.5. Adding A-Tail

The inclusion of a segmented poly(A) tail with spacer elements (usually short—about 10 nucleotides) in the DNA template can also improve the translation efficiency and half-life of mRNA [90].

### 4.2. Ionizable Lipids

Ionizable lipids play a very important role in the formation of LNPs and in the transfection process in vivo. Firstly, ionizable lipids are electrostatically complexed with negatively charged mRNA molecules to form a complex that improves the stability of the mRNA molecules. The LNPs are neutral in physiological pH due to ionizable lipids and PEG lipids, thus reducing the non-specific interactions with serum proteins. Then, after the dissociation of the PEG lipids, cells take up the LNPs via the apolipoprotein E (ApoE)-dependent pathway, and when LNP@mRNA reaches the cell membrane, the ionizable lipids trigger membrane fusion with the negatively charged cell membrane and are taken up by the cells through endocytosis. Finally, as lysosomes containing a variety of hydrolytic enzymes break down exogenous macromolecules, the pH decreases, thereby creating an acidic environment. When protonated at a low pH, ionized lipids can induce the formation of hexagonal phase structures that disrupt the bilayer structure of the LNPs, thus allowing the mRNA to be released into the cell, where it binds to the ribosomes responsible for protein production according to the “central law” and is translated into viral proteins (antibodies) that neutralize the virus (Figure 5).

In the early stages of vaccine research, researchers used positively charged cationic lipids to interact with negatively charged mRNA for the purpose of delivery. However, as the cationic lipids used in the early days were also positively charged under physiological conditions, they were prone to not only interact with other negatively charged molecules in biological fluids, but also to be trapped by immune cells. At the same time, cationic lipids are often cytotoxic, so the in vivo safety of cationic lipids had to be considered in the formulation design. In addition, positively charged LNPs are easily and rapidly removed from the circulation due to the adsorption and uptake of serum proteins by the RES [91]. Positive charges also usually increase the cytotoxicity of a given system [92], which limits their value for application. Hence, researchers later developed ionizable lipids with a changing polarity via pH (lipids or lipids containing amine groups are ionizable lipids) as the core technology of LNPs, whose molecular structure determines the targeting, delivery efficiency, and stability of the formulation and the metabolic kinetics and toxicity of LNPs.

#### 4.2.1. The Effects of Ionizable Lipid Structure on the Efficiency of LNP Transfection

The structure of the ionizable lipid can be divided into three parts: head, connecting fragment, and tail (Figure 6).

##### Head

The head groups of ionizable lipids are usually positively charged. The size and charge density of the head group are mainly involved in processes of wrapping nucleic acids, stabilizing LNPs, interacting with cell membranes, and facilitating endosomal escape. Common ionizable lipids usually contain only one head group but sometimes several head groups. Typical head groups include the amine (primary amine, secondary amine, tertiary amine, and quaternary amine), guanidine, and heterocyclic groups. The clinically used ionizable lipids (DLin-MC3-DMA((6Z,9Z,28Z,31Z)-heptatriaconta-6,9,28,31-tetraen-19-yl 4-(dimethylamino)butanoate), SM-102(heptadecan-9-yl8-((2-hydroxyethyl)(6-oxO-6-(undecyloxy)hexyl)amino) octanoate), and ALC-0315((4-hydroxybutyl)azanediyl)bis(hexane-6,1-diyl)bis(2-hexyldecanoate)) contain tertiary amine heads that undergo pH-dependent ionization. The head groups of ALC-0315 and SM-102 also contain a terminal hydroxyl group that reduces hydration of the head group and improves hydrogen-bonding interactions with nucleic acids, thereby potentially improving the transfection capacity.

##### Connecting Fragment

The connecting fragment can connect the head to the tail and sometimes also hides within the tail (SM-102 and ALC-0315), which affects the stability, biodegradability, cytotoxicity, and transfection efficiency of LNPs. Ionizable lipids may contain one or more linker fragments, while most ionizable lipids contain only one type of linker fragment, probably due to the easy synthesis. Conjugates can be divided into non-biodegradable (ethers and carbamates) and biodegradable (esters, amides, and thiols). Biodegradable linking fragments are preferred because they are usually cleared rapidly in vivo, which allows for multiple doses and reduced side effects. DLin-MC3-DMA, ALC-0315, and SM-102 all contain ester-bonded linking fragments. Of these, modifications around the ester group of SM-102 affect LNP clearance, formulation stability, and transfection efficiency.

##### Tail

The hydrophobic tail affects the p*K*a, lipophilicity, mobility, and fusion, thus affecting the formation and efficacy of LNPs. Ionizable lipids generally contain one to four hydrophobic tails containing 8 to 20 carbon atoms. The tails can be saturated or unsaturated lipid chains whose degree of unsaturation can affect the nucleic acid delivery by modulating the properties associated with membrane instability. DLin-MC3-DMA has two linoleic tails, while ALC0315 and SM-102 have two branching saturated tails that are thought to have a tapered geometry and promote endosomal membrane instability and intracytoplasmic release of nucleic acids.

#### 4.2.2. Key Points in the Design of Chemical Structures for Ionizable Lipids

Most of the main components currently used in the preparation of LNPs are ionizable lipids that have the ability to modulate the p*K*a, which can increase the loading rate of the nucleic acids and further enhance the effect of gene therapy; when mixed with polyanionic nucleic acids, they can self-assemble to form LNPs. In the preparation of LNPs, lower-pH conditions will result in positively charged ionizable cationic lipids, which leads to higher nucleic acid loading. During drug delivery, when the pH is higher than that of the physiological environment of the ionizable cationic lipid p*K*a, the LNP surface presents an almost neutral charge that can escape RES uptake, prolong in vivo circulation, and reduce its cytotoxicity. Once an LNP enters the intracellular endosome (where the pH is lower than the p*K*a of the lipid), the amine group of the ionizable cationic lipid protonates and binds to the anionic group on the endosome membrane, thus facilitating the escape of the nucleic acid from the endosome [93].

##### p*K*a

A key characteristic of an ionizable lipid is the p*K*a, which is the pH at which 50% of the ionizable lipid in an LNP is protonated as measured by a TNS dye-binding assay (generally ranging within 6–7). At a low pH, the positive charge forms a complex with the negatively charged mRNA molecules, thus protecting them from degradation and stabilization. At a neutral pH, it protects the structural integrity of the LNP and reduces the incidence of toxic side effects. When lipid nanoparticles are endocytosed, the lower pH in the endosomes results in the lipids being protonated and positively charged. Then, the stability of the membrane structure becomes less stable or is even destroyed, thereby facilitating the uptake of the LNP by the cells and the escape of the endosomes.

##### Endosome Escape

Although endosomes can help LNPs enter the cell, a large amount of LNPs may be constrained within the endosome, which results in the mRNA being destroyed by the endosome before release. Therefore, the development of better ionizable lipids is key to improving the efficiency of LNP@mRNA delivery. According to the molecular shape hypothesis, the conical shape of the ionizable lipid tail is larger than the cross-section of its head group; endosome escape is another key feature of ionizable lipids. In response to the increase in branching, Acuitas designed LipidA9 with five branching chains, which produced ionizable lipids with a more conical structure and therefore greater membrane disruption when paired with the anionic phospholipids in the endosome [94,95]. Lipid ALC-0315 introduced more branches in the alkyl tail compared to lipid MC3 to strengthen its tapered morphology and thus enhance the endosomal release of LNPs. In addition, Huang et al. [96] designed and studied the novel ionizable lipid-like molecule iBL0104, which protonates in an acidic environment, thereby leading to an increase in the cationic groups and causing the formation of ion pairs between the cationic lipid and the anionic lipid of the endosome/lysosome membrane. This ion pair takes the shape of a “molecular cone” and changes the membrane phase of the endosome/lysosome, thereby forming an antiphase non-bilayer phase, thus rendering the membrane of the endosome/lysosome unstable and enabling the efficient escape and release of the siRNA.

#### 4.2.3. Progress in the Study of Ionizable Lipids

Lipids based on MC3 were initially developed for the delivery of mRNA molecules. The key cofactor for the world’s first RNAi drug (Onpattro™) was the cationic lipid material DLin-MC3-DMA, which has a unique pH-dependent charge-variable property: it is positively charged under acidic conditions and electrically neutral under physiological pH conditions, which effectively encapsulates the corresponding siRNAs and allows them to enter the cytoplasm for efficacy. However, since siRNA products require repeated administration to treat chronic diseases, the slow degradation of the alkyl dioleate tail in MC3 may lead to accumulation, thereby resulting in potential toxicity when repeatedly administered. Therefore, degradable lipids were later introduced into MC3 lipids by replacing one of the two double bonds in each alkyl chain with a primary ester that could be readily degraded by esterases in vivo, which resulted in the development of L319 lipids with a higher delivery efficiency and faster elimination from the liver and plasma without altering their delivery rate of mRNA molecules. L319 was shown to have a half-life of less than one hour in the liver and maintained a similar efficiency of gene silencing to that of MC3 in the liver [97]. Similarly, SM-102 and ALC-0315, both of which had similar design principles and structurally similar functional groups, were developed for Moderna and BioNTech’s new crown vaccines (mRNA1273 and BNT162b2, respectively) to optimize the delivery efficiency and pharmacokinetics. Compared to the structure of MC3, the structures of ALC-0315 and SM-102 have the following main changes: (1) the hydrophobic chain does not contain unsaturated C=C, thus making LNPs less susceptible to oxidation and more stable; (2) the introduction of ester bonds in hydrophobic chains enhances hydrophobic interactions with endosomal membranes, increases the fusibility of endosomal membranes and improves escape efficiency, accelerates tissue clearance, reduces toxicity, and improves tolerance while being easily degraded by esterases; and (3) increased hydrophobic chain branching enhances LNP ionization in acidic endosomes, which leads to an increased endosome escape efficiency and increased mRNA protein expression.

In addition to the lipid structure, the proportion of ionizable lipids in LNP lipid compositions is critical in determining the encapsulation and delivery efficiency of LNP systems. LNPs consisting of ionizable cationic lipids, phospholipids, cholesterol, and PEG lipids in a ratio of 20:25:45:10 (mol%) showed a high retention efficiency but were less effective in gene silencing in hepatocytes [98]. Increasing the molar ratio of ionizable lipids to 40 mol% could enhance hepatic accumulation and gene-silencing activity [99], and a molar ratio of lipid composition of 50:10:38.5:1.5 mol% resulted in a 6-fold improvement in hepatocyte gene silencing [100].

In 2008, Akinc et al. [101] reported for the first time a subset of ionizable lipids called “lipid-like” substances, which are ionizable lipid derivatives composed of tertiary amines that are mainly used for LNP-mediated siRNA delivery. Significant progress has been made in the development of “next generation lipids and lipid-like molecules” for LNP@mRNA delivery systems [45,102]. Compared to MC3, the new aminolipid series was optimized to maintain immune titration while providing better biodegradability along with a significantly improved tolerance in rats and non-human primates. Compared with typical ionizable lipids, subsequent lipid analogues have shown significant improvements in potency and overall performance; for example, C12-200, cKK-E12, of-02, and 503-O13 require much lower doses. Anderson et al. [103] optimized an LNP@mRNA formulation consisting of C12-200 and DOPE with an adjusted ratio of mRNA to C12-200 of 1/10 (*w*/*w*). Compared with LNPs containing DLin-MC3-DMA, the preparation successfully increased the level of production of translated proteins with both systems predominantly distributed in the liver. Cationic lipid-modified aminoglycosides (CLAs), which are produced from naturally occurring aminoglycosides modified with alkyl epoxides and acrylates, also recently were explored as candidates for the cationic component of the LNP@mRNA system. Yu et al. [104] generated libraries of CLAs paired with a DOPE/cholesterol/DMG-PEG system to deliver EPO mRNA in mice and demonstrated that the blood levels of EPO were more than 3-fold higher 6 h after administration of LNPs containing GT-EP10 (one of their drug candidates) compared to a commercially available MC3-based formulation.

Overall, the design, synthesis, and screening of efficient novel ionizable lipids is a key factor in the rapid development of mRNA technology; advances in ionizable lipid formulations will ensure that LNP@mRNA systems can maximize the encapsulation efficiency without sacrificing the biocompatibility. In view of this, Michael J. Mitchell’s team [105] classified five types of ionizable lipids for mRNA delivery based on their structural properties: unsaturated (containing unsaturated bonds), multi-tailed (containing more than two tails), polymeric (containing polymer or dendritic macromolecules), biodegradable (containing biodegradable bonds), and branched (containing branching tails) lipids. They summarized the various ionizable lipids as follows: (1) the saturation of the tails of ionizable lipids greatly affected the mobility and delivery efficiency of ionizable lipids; (2) the increased cross-section of the tail region of the multi-tailed ionizable lipids could produce more tapered structures with greater endosomal disruption; (3) ionizable polymer–lipids often contain mixtures of different substituted compounds, thereby increasing the compositional complexity; (4) the introduction of biodegradable chemical bonds in ionizable lipids can reduce the toxicity of ionizable polymer–lipids by introducing biodegradable chemical bonds (a common strategy is to introduce ester bonds that are stable at physiological pH but are hydrolyzed by enzymes in tissues and cells); and (5) with an increasing tail length and saturation, tail branching greatly affects the performance of ionizable lipids. Ionizable lipids containing isodecyl acrylate significantly increased liver mRNA expression compared to isomers with linear tails. Through these generalized classifications, they also pointed out the current state of development and the problems that exist, which in turn will guide the development of the next generation of ionizable lipids to better serve the delivery of mRNA drugs.

Miao et al. [106] developed a combinatorial library of ionizable lipids to identify mRNA delivery vectors that facilitated mRNA delivery in vivo and provided effective and specific immune activation. The best-performing lipids were found to share a common structure: unsaturated lipid tails, dihydroimidazole junctions, and cyclic amine head groups as measured using a three-dimensional multi-component reaction system. These agents enhanced the anti-tumor effect not by activating Toll-like receptors, but by inducing antigen-presenting cell maturation through activation of the intracellular STING signal pathway. Among them, the preferred agents induced a robust immune response and were able to inhibit tumor growth in melanoma and human papillomavirus E7 tumor models and prolonged the survival of tumor-bearing mice.

Recently, Dong et al. [79] summarized their recent progress in the design and development of various classes of lipids and lipid derivatives that can be formulated using many types of mRNA molecules to treat various diseases. The group conceived a series of ionizable lipid-like molecules based on a benzene core, amide linker, and hydrophobic tail structure, and they identified N1, N3, N5-tris(3-(dilaurylamino) propyl) benzene-1,3,5-tricarboxamide (TT3) as a lead compound for in vitro and in vivo mRNA delivery. The biodegradability of these lipid-like molecules was tuned by the introduction of branched or linear ester chains. Inspired by bionic compounds, many classes of ionizable lipids have been synthesized, including lipid-like molecules, biodegradable lipids, aminolipids derived from chemotherapy drugs, bionic phospholipids and glycolipids, and vitamin-derived ionizable lipids, which greatly broaden the chemical space of ionizable lipids for mRNA delivery. In addition, Li et al. designed and constructed various ionizable lipids by building an in-house LNP technology platform [106]. In the face of different application scenarios, the synthesized LNP structures and activity data were analyzed and optimized to simulate the delivery and therapeutic efficacy of reduced nucleic acid drugs via high-throughput screening and to screen for the best-in-class LNP structures that met specific needs. The authors claimed that they built a library of nearly 5000 LNPs to screen LNP vectors for different therapeutic scenarios [107].

### 4.3. PEGylated Lipids

As another important LNP component, PEGylated lipids have multiple roles in reducing the interaction of LNPs with serum proteins as well as in reducing the aggregation or fusion of LNPs and increasing the circulation time and stability of LNPs in vivo. Consisting of two structural domains (a hydrophilic PEG polymer bound to a hydrophobic lipid anchor), PEGylated lipids are located on the surface of lipid particles with the lipid structural domains buried deep within the particles and the PEG structural domains extending from the surface. They have been applied in the liposome system to prolong the circulation time because PEG polymers represent a spatial barrier that prevents the binding of plasma proteins that would otherwise lead to their rapid clearance by the RES. Due to the enhanced permeability and retention (EPR) effect, the longer the particles circulate in the bloodstream, the more likely they are to accumulate at disease sites such as solid tumors [108,109]. The content of PEGylated lipids is typically 0.5–5%, with DMG-PEG2000 in both the Moderna new crown mRNA vaccine and DMG-C-PEG2000 in the SiRNA drug Onpattro™ at 1.5% and ALC-0159 in both the Pfizer-BioNTech new crown mRNA vaccines at 1.6%.

Therefore, different PEGylated lipids control the circulation time of LNPs in vivo and influence the interaction of LNPs with cells. The choice of the PEGylated lipid depends primarily on the therapeutic target, the target organ, and the route of administration; the molar ratio and length of the hydrophobic chain portion of the PEGylated lipid must also be considered.

### 4.4. Auxiliary Phospholipids

Auxiliary lipids are mainly utilized in the study of liposomes to enhance the stability and improve the circulation in vivo. Auxiliary lipids in LNP formulations for nucleic acid delivery are a class of neutral lipids that are distinct from cationic lipids and include sterols, phospholipids, and glycerolipids. They are usually derived from conventional liposomal auxiliary lipid formulations and have similar roles in LNPs to those in traditional liposomes.

Several studies have shown that the properties of phospholipids may strongly influence the efficacy of LNP@mRNA [103]. Currently, the construction of LNP@mRNA commonly uses phosphatidylcholine (PC), 1,2-distearoyl-sn-glycero-3-phosphocholine (DSPC), and 1,2-dioleoyl-sn-glycero-3-phosphoethanolamine (DOPE) as co-lipids; these display high melting temperatures and enhance the stability of LNP formulations by forming bilayer structures. However, among these, DSPC and DOPE may present significant differences, and it has been reported that replacing DOPE with DSPC for mRNA in LNP preparations either shows an improvement [47] or does not work well [110]. Kauffman et al. [103,111] noted that although their formulation of LNPs with DSPC had a better rate of mRNA encapsulation compared to the DOPE-containing formulation, the latter induced better protein expression. In terms of safety, Oberli et al. [47] found that in 20% of mice, LNPs formulated with DSPC or DOPE caused inflammation at the injection site 5–10 days after administration.

The current study of the relationship between auxiliary and ionizable lipids is also key to improving LNP@mRNA. Recent studies showed that increasing the molar ratio of amphiphilic phospholipids to ionizable cationic lipids improved mRNA delivery [112] and that synthesis of a new generation of ionizable phospholipids could reduce the amount of cationic lipids while contributing to mRNA encapsulation [113].

### 4.5. Cholesterol

Cholesterol is a component of cell membranes; when combined with phospholipids in the traditional bilayer structure of liposomes, cholesterol promotes the formation of a bilayer structure with a low membrane fluidity and an increased thickness. Studies have confirmed that cholesterol is no longer in a liquid form in LNP formulations that deliver nucleic acids and that high molar levels of cholesterol lead to the formation of insoluble cholesterol microcrystals in the core of an LNP, which leads to the “deprotonation” of cationic lipids. As with ionizable lipids, cholesterol is present on the surface of LNPs in a crystalline form, thus affecting the ability of LNPs to escape from endosomes [114]. Sahay et al. [66] found that different types of cholesterol improved the rate of LNP escape from lysosomes. A recent study on cholesterol analogues showed that the introduction of C-24 alkyl phytosterols into LNPs, which were termed as eLNPs, could enhance gene transfection. The length of the alkyl chain, the flexibility of the sterol ring, and the polarity of the hydroxyl group were important factors that influenced the transfection capacity. Compared with the spherical structure of a conventional LNP, the structure of an eLNP is polyhedral; eLNPs also could enhance the cellular uptake and retention and had a greater capacity for stable release from endosomes; however, these experiments were conducted in vitro, so the application to humans still needs to be validated [66].

### 4.6. Particle Size of LNPs

The particle size of nanoparticles may be attributed to the composition (lipid, surfactant and dispersion medium) and homogenization parameters. Increasing the proportion of the surfactant, raising the homogenizing pressure and temperature, and prolonging the homogenizing time can reduce the particle size. Lipid viscosity is also an important parameter that affects the particle size. Nanoparticles with a hydrodynamic diameter of less than 5 nm are usually rapidly cleared by renal filtration after intravenous administration [115]. Nanoparticles between 15 and 200 nm in size often accumulate in the liver and spleen after being trapped by Kupper cells and macrophages, while particles larger than 200 nm accumulate in the spleen [116,117]. Meanwhile, PEGylated lipids control the size of LNPs by providing a hydrophilic shell that limits vesicle fusion during assembly; studies have shown that appropriate increases in the concentration of PEGylated lipids produce smaller LNPs [118]. When the molar fraction of PEGylated lipids was changed from 0.25% to 5%, the LNP size was reduced from 117 nm to 25 nm.

Basha et al. [119] tested the efficiency of gene transfection in mouse hepatocytes and hepatic macrophages using LNPs with a particle size of 80–360 nm and concluded that the delivery within the liver was more favorable with smaller particles, while the opposite was true for macrophages. However, this conclusion was refuted in a subsequent study in which LNPs that were 27–117 nm in size were prepared by varying the amount of PEG lipids (0.25–5%) and assessed for hepatic gene-silencing activity after intravenous injection into mice [120]. All of the LNPs showed a similar accumulation in the liver; the optimal LNP size for silencing hepatocytes was 78 nm. Both the smallest (27 nm) and largest particles (117 nm) showed a poor efficacy, which may have been due to the stability of the LNPs (rapid breakdown of the lipid component) or their inability to penetrate the dense blood vessels of the mouse liver. Later, they changed the PEG lipid content of the LNPs (0.25–3%) and prepared LNP@mRNA containing human erythropoietin (hEPO) at 45–135 nm [121]. However, there were only slight differences in the cellular uptake for different LNP particle sizes; 65 nm LNPs expressed the highest hEPO in hepatocytes and adipocytes in vitro.

Additionally, Moderna’s study showed that although LNP particle sizes of around 100 nm produced consistently high antibody titers in mouse immunization experiments, which meant a lower immunogenicity, curiously, in non-human primates all of the tested LNPs produced robust immune responses at particle sizes of 60–150 nm [122]. These studies demonstrated the important effect of the particle size on efficacy. However, as mentioned above, changing the percentage content of PEG lipids had a strong effect on the course of the particles in vivo, in which introduced an additional variable for screening. Overall, these conflicting results regarding the optimal LNP size suggest that multiple factors may contribute to efficient LNP-mediated RNA delivery, including the LNP composition, PEG lipid content, etc. Therefore, further mechanistic studies are needed to elucidate this size–activity relationship.

## 5. The Factors Affecting the Distribution of LNP@mRNA In Vivo

### 5.1. Route of Administration

The route of administration is one of the most important factors that affects the distribution of LNPs. Intramuscular injection is commonly applied for vaccines because it contributes to lymph node targeting and activation of the immune response. When administering the vaccine, antigen-presenting cells are recruited to the point of delivery where they can encounter the vaccine antigen. Then, they are transferred to the lymph nodes and stimulate a T-cell response. LNPs administered intravenously are mainly distributed in the liver and spleen but also in the lungs. LNPs with net positive, neutral, and negative charges can be used to target the lung, liver, and spleen, respectively. The addition of cholesterol or polyethylene glycolyzed lipids to the formulation, as well as increasing the size of the LNPs, increase the distribution to the spleen. It is worth noting that formulations that are optimized for a particular route of administration are usually not applicable to other routes of administration. ApoE in serum binds to LNPs and is also predominantly cleared by the liver, which leads to mRNA entering the liver preferentially and thereby increasing the hepatic targeting of LNPs and mitigating its toxicity, but it is clear that the full potential of LNP@mRNA cannot be realized. Therefore, a method to achieve selective delivery outside the liver is currently one of the main challenges facing LNP@mRNA.

### 5.2. Targeted Molecular Modification

Targeted molecular modification on the surface of LNPs is the most direct way to achieve targeted delivery to cells. The principle is to add peptides, antibodies, or proteins that target specific cell surface molecules to the nanoparticles that carry the mRNA to prevent liver accumulation and achieve specific transport of the mRNA to target cells through high-affinity binding. Earlier, scientists coupled eight different monoclonal antibodies to the surface of LNPs, thereby allowing siRNA to be specifically taken up by different leukocyte subpopulations of the siRNA [123]. It also was shown that mannosylation of lipid nanoparticles (LNPs) (LNP-MAN) could potentially enhance uptake of antigen-presenting cells; LNP-MAN also had a superior gene-delivery efficiency in vitro and in vivo to that of LNPs [124]. In addition, by modulating the surface charge of RNA–lipid complexes (RNA-LPX), it was also possible to target dendritic cells (DCs) precisely and effectively in vivo [87]; this method has now been used to improve cancer vaccines. For mRNA vaccines, the translocated antigen is taken up by antigen-presenting cells and transported after administration to the germinal center, where it produces a high-affinity neutralizing antibody against the antigen. Therefore, if LNP@mRNA can target antigen-presenting cells, it will undoubtedly enhance the application of mRNA technology.

Rurik et al. [125] chose this strategy for their study on the treatment of cardiac injury with chimeric antigen receptor (CAR)-T cells in vivo by coupling a CD5 antibody to an LNP (CD5/LNP) to deliver CAR mRNA that encoded a fibroblast-activating protein (FAP). The LNP vector was able to target splenic T cells in mice that highly expressed CD5 by specifically targeting T cells and generating FAP-(CAR)-T cells that recognized cardiac fibroblasts, thereby reducing fibrosis and restoring cardiac function in a mouse model of heart failure, which greatly expanded the application prospects of LNP vectors and (CAR)-T. Peer et al. [126] demonstrated potential therapeutic promise in a mouse model of colitis with an LNP-modification technique via a recombinant fusion protein that recognized the specific protein conformation of integrin α_4_β_7_ expressed on intestinal leukocytes and selectively delivered mRNA to specific leukocyte subpopulations. Recently, LNPs successfully targeted brain regions of mice with acute brain inflammation and attenuated TNF-α-induced brain edema using an approach that targeted vascular cell adhesion molecule-1 ligands [127]. In addition, the laboratory coupled nucleoside-modified LNP@mRNA to the platelet endothelial cell adhesion molecule-1 to redirect LNP accumulation from the liver to the lungs during brain edema [128]. In a method unlike antibody modification, Gabriel et al. [129] recently developed a light-controlled MHC class I antigenic-peptide ligand-replacement platform that rapidly transforms the targeting antigenic peptides modified to the surface of LNPs to enable targeted mRNA delivery to multiple antigen-specific T-cell populations in vivo. Hence, it was revealed that targeted molecular modifications play a significant role in affecting the in vivo distribution of LNP@mRNA.

### 5.3. High-Throughput Screening and Design of Predictable LNP

High-throughput screening through structurally diverse chemical libraries has long been proven to be one of the most effective methods for the development of targeted LNPs; most LNPs are currently developed using this technique, but the method is somewhat blind and requires a very large amount of work, which is very worrying in terms of efficiency. Dahlman et al. [130] developed a novel high-throughput screening technique known as FIND that combines “DNA barcoding” and deep sequencing to significantly improve the efficiency of LNP screening. By wrapping Cre mRNA and sequence-specific DNA barcodes into the same LNP, they were able to create a series of LNP@mRNA drugs with specific “tags” by simply designing different DNA barcodes. These differently labelled LNPs were injected intravenously into Cre-induced tdTom-expressing transgenic mice; the tdTom-positive cells were detected using flow cytometry combined with deep DNA sequencing to identify the specific cell-targeting LNP vectors. With this system, they could screen over 200 LNPs simultaneously, which significantly improved the efficiency of LNP screening.

In contrast to high-throughput screening, if clear principles for the design of LNPs are given, the development cycle is significantly reduced and the efficiency of the development is improved. Siegwart et al. [131] achieved a targeted delivery to the lung, spleen, and liver by adding only the SORT molecule to a four-component LNP. As shown in Figure 7, the addition of permanent cationic selective organ targeting (SORT) lipids (DDAB, EPC, and DOTAP) transferred tissue tropism from the liver to the lung. The addition of permanent anionic SORT lipids (14PA, 18BMP, and 18PA) achieved specific delivery to the spleen, and appropriate ionizable cationic SORT lipids (DODAP and C12-200) increased liver targeting. It was also exciting to note that the SORT technology is universally applicable and allows for the rapid development of extrahepatic organ-targeted LNPs based on multiple types of LNPs. However, further mechanistic exploration revealed [132] that SORT molecules regulated the biodistribution of SORT-LNP at the organ level, LNP’s own p*k*a, and the surface serum protein crown, which ultimately determined the ability of the LNPs to be used for delivery in vivo. It is worth noting that these studies were conducted via intravenous administration and that the optimization for this route is not usually applicable to the current mainstream intramuscular route.

### 5.4. Discovery of Novel Ionizable Lipids

As an important component of LNPs, ionizable lipids significantly influence the distribution of LNPs in vivo. To achieve the transport and aggregation of LNPs at specific sites, many novel ionizable lipids have been discovered and applied. Liu et al. produced a library of 51 ionizable iPhos phospholipids that were used to formulate LNP@mRNAs of a similar size, zeta potential, and p*K*a but that displayed a range of in vivo efficacy and organ selectivity [113]. After intravenous administration of mRNA at 0.1 mg/kg, the iPhos lipid-containing LNP, which consisted of a single tertiary amine and phosphate group and three alkyl tails, showed the highest efficacy; the alkyl chain length played a key role in the organ selectivity and functional mRNA delivery. On the amine side, 8–10 carbons promoted high mRNA expression. On the phosphate side, the translation was directed to the liver, where chains were below 12 carbons, while longer chains were confined to the spleen. However, the discrete pathways for tuning the physicochemical properties of formulation components to guide the selective transport and expression of LNPs in certain organs and tissues ultimately require more detailed mechanistic studies to establish.

Fenton et al. [133] synthesized a novel lipid (OF-Deg-Lin) for selective mRNA delivery in the spleen and demonstrated that the LNP based on OF-Deg-Lin could specifically produce proteins in B cells; this was an ionizable lipid containing degradable linkers. They prepared Cy5-labelled, mRNA-loaded LNPs and assessed their biodistribution and luciferase expression, which revealed that the OF-Deg-Lin-based LNPs were predominantly delivered to the liver.

In a recent study, Xu et al. [134] found that synthetic imidazole-based lipid compounds preferentially targeted mRNA to the spleen. Meanwhile, Qiu et al. and Xu et al. [135] discovered that the N-series of these LNPs (tail structures containing amide bonds) were able to specifically deliver mRNA to mouse lungs by screening of their synthetic lipid nanoparticle libraries in vivo. Furthermore, they found that by simply changing the structure of the ionizable head, which meant changing the intermediate linking group of the synthetic lipid compound molecule from an ester bond (O series) to an amide bond (N series), the organ targeting of the LNPs could be changed from the liver to the lung, thereby targeting the delivery of the mRNA to different lung sub-cell types. The team developed a lung-targeted LNP to specifically deliver mRNA that encoded the normal Tsc2 gene into Tsc2-deficient TTJ cells for the treatment of lymphangioleiomyomatosis (LAM) caused by mutations in the Tsc2 gene. The experiment demonstrated the success of LNP-based mRNA therapy in a preclinical model of LAM, which suggested that LNP@mRNA is a promising therapeutic intervention for LAM. Meanwhile, the team was able to deliver the CIRSPR-Cas9 gene-editing system to the lung, but the specificity was not yet adequate, and a small fraction was still taken up by the liver. This may be mainly because co-encapsulation of Cas9 mRNA and sgRNA affected the physicochemical properties of the LNPs to a certain extent, which in turn affected their lung targeting. Additionally, it is worth noting that in this study, the researchers found that when LNPs were injected into the bloodstream, they specifically adsorbed proteins from the plasma and formed a “protein crown” on their surfaces, which then targeted the LNPs to specific tissues, organs, or cells. They found that the composition of the protein crown of the LNPs could be adjusted via simple chemical methods to regulate the in vivo organ targeting of the LNPs, which provides new ideas for the rational design of organ- and cell-selective LNP@mRNAs with a high specificity and that in the future could even target other organs such as the heart, brain and pancreas. In addition, by analyzing the protein crown components on the surface of liver-targeting and lung-targeting LNPs, the scientists identified 14 proteins that may influence the lung targeting of LNPs, including albumin, ApoE, fibrinogen β, and fibrinogen γ. It remains to be further investigated which protein or proteins play the decisive role. In addition, changes in lipid species or alternative approaches such as LNPs that contain bound phospholipids [136] and engineered LNPs [137] can also alter the targeting of LNPs.

In the future, achieving targeted delivery of mRNA will become more challenging and critical, which may accelerate the widespread use of mRNA technology in precision medicine. The combination of tissue- and cell-targeting strategies may be a good method, but there are still some issues to be addressed. For example, ligand/antibody-modified LNPs may disrupt the balance of internal components and the recruitment of protein crowns in the blood, thus affecting the established tissue specificity of the LNPs. We believe that with further research on targeted delivery, more-precise targeting strategies will emerge and be applied in precision medicine in the future.

## 6. Problems in Clinical Application of LNP@mRNA

### 6.1. Immune-Related Adverse Effects

Currently, while mRNA vaccines continue to be administered, allergic reactions to them keep being reported. Of these, the incidence of severe allergic reactions was higher in those who received BNT162b2, while less severe allergic reactions were reported in those who received the mRNA-1273 vaccine [138]. Gurudeeban Selvaraj et al. concluded that the causes of allergic reactions to mRNA vaccines were mainly allergens present in the glycoprotein (amino acid residues 437–508 sequence of spinosin) or stabilizers (sucrose and PEG) of the vaccine [139]. These molecules promote or stimulate allergen-specific antibodies (type E or IgE immunoglobulins) in the body, which react with IgE and cause an allergic inflammatory response [140]. Allergens can further lead to the release of inflammatory mediators (histamine, prostaglandins and leukotrienes, and proteases) and pro-inflammatory cytokines, producing allergic reactions such as nausea, vomiting, redness, rash, laryngeal oedema, wheezing, tachycardia, hypotension, and cardiovascular collapse [140].

Although synthetic modification of mRNA significantly reduces its associated immunostimulatory effects and significantly increases the immunogenicity of LNPs [141], it is possible that immune activation and cytotoxicity can be triggered when more than a certain dose of mRNA or some ionizable lipid is given in the form of LNPs. Nucleoside-modified mRNA translates into a SARS-CoV-2 spike-in glycoprotein fragment, which may contain a water-soluble glycoprotein fragment (allergen) that triggers an allergic reaction in some vaccinated individuals. The innate immune system is activated when phagocytes of the RES recognize the lipid component of LNP. Once activated, the Toll-like surface receptors (TLR2 and TLR4) can trigger the induction of high levels of cytokines, which is called cytokine release syndrome [142]. Abrams et al. [143] studied this phenomenon after the systemic administration of LNPs consisting of CLinDMA (2-{4-[(3β)-cholest-5-en-3-yloxy]butoxy}-N,N-dimethyl-3-[(9Z,12Z)-octadeca-9,12-dien-1-yloxy] propan-1-amine) and found that they produced both pro- and anti-inflammatory cytokines. These were mainly related to the lipid content of the LNPs and their payload.

Interleukin-1 (IL-1) is essential for boosting the body’s immune system. IL-1 would trigger the release of broad-spectrum pro-inflammatory factors, including IL-6, but this inflammatory factor may also induce certain side effects. SiriTahtinen et al. [144] found that the ionizable lipid SM-102 significantly activated this pathway, which may be the reason why Moderna’s mRNA vaccine exerted a stronger protective effect while causing nausea in some vaccinees. Such lipid molecules trigger a range of inflammatory molecules, which both promote efficacy and cause adverse reactions. Researchers expect to design ionizable lipids that activate favorable immune responses without over-stimulating harmful immune pathways. In addition, LNP may also induce serum complement activation, leading to a non-IgE mediated hypersensitivity reaction known as complement activation-related pseudo-allergy, ultimately leading to anaphylaxis [145,146].

To address the immune stimulation caused by LNP@mRNA, major clinical studies have focused on the use of the glucocorticoid receptor (GR) agonist dexamethasone [143] as well as corticosteroids to reduce the immune stimulation of mRNA [147]. In addition, Tao et al. [148] further showed that Janus kinase inhibitors effectively attenuated LNP-siRNA-induced toxicity while significantly suppressing all associated toxic responses (cytokine induction and aspartate aminotransferase and alanine transaminase (ALT and AST) elevation). This approach may be more beneficial than corticosteroids with multiple immunosuppressive properties.

### 6.2. Side Effects of PEG

PEGylated lipids are not sufficiently stable because the PEGylated lipid has 14 carbon atoms in its alkyl tail, so it is not stably anchored to the LNP surface and has been found to gradually shed the ionizable lipids MC3 and DSPC from the LNPs during circulation. The shedding of the PEG is thought to enhance LNP transfection at some point, but if too extreme, it leads to a rapid loss in ionizable lipids and DSPC, which would negatively affect endosomal release.

PEG inhibits cellular uptake and transfection efficiency. Due to the spatial site-blocking effect of the PEG chain, it shields LNPs from interactions with cell membranes and inhibits their uptake by target cells, while the PEG–lipid structure plays an important role in the circulating half-life: particles with longer circulation times can lead to the production of anti-peg IgM and IgG by splenic B cells, leading to a loss in potency after repeated LNP administration due to the accelerated blood clearance (ABC) effect [149]. LNP@mRNA [56] has been shown to produce anti-PEG antibodies in mice and is currently addressed mainly by using optimized PEG–lipid fractions in terms of the density and chain length, which is essential for the efficient development of LNP@mRNA systems. Breakable PEG–lipid linkages include ester, hydrazone, peptide, or short-chain lipids. The application of C14 lipid anchoring makes it easier to dissociate PEG from the particle surface than that of C18. The “diffusible” PEG lipid consists of a C14 acyl chain lipid that balances the transfection efficacy and stability of the LNP system. It dissociates rapidly from the LNP surface into the circulation after administration, accumulates in the liver, allows cellular uptake of particles, and reduces PEG-mediated antibody responses; the anti-PEG antibody production is reduced by a factor of 10 compared to the use of more stable monolithic (C16 or C18) PEG liposomes [120,150]. In addition, the “persistent” PEG lipids consist of C18 acyl chains, which are more effective in preventing premature clearance of LNPs, and formulations incorporating C18PEG lipids at molar ratios in excess of 1.5% increase the LNP cycle time, thereby enhancing the accumulation of the LNP system in the target tissue and targeting extrahepatic without loss of transfection potency [151]. In addition, to address the PEG dilemma, Heinrich Haas et al. employed pSarcosinylated lipids as a tool for particle engineering; the pSarcosinylated nanocarriers without a compromised transfection efficiency showed less secretion of pro-inflammatory cytokines and a reduced activation of complement compared to the PEGylated counterparts [152].

PEG causes allergic reactions. Studies have shown that PEG is also responsible for allergic reactions to the Pfizer/BioNTech mRNA COVID-19 vaccine and that the trigger for allergic reactions is thought to be due to the IgE of anti-PEGylated lipids. Several incidences of acute allergic reactions have been observed since the current emergency use approval of Pfizer’s mRNA COVID-19 vaccine [153]; these were approximately 10 times higher than for the other vaccines. The source of this allergic reaction may be the production of antibodies to PEG in the body. In recent years, an increasing number of biological drugs also have contained PEG, and more people are developing antibodies against PEG; this may be caused by the use of cosmetics, daily products, or medicines containing PEG.

### 6.3. Instability Issues of LNP@mRNA

The exposure of the mRNA to the water at the core of an LNP makes the mRNA very vulnerable; the LNP colloidal stability is also critical to the quality of the LNP@mRNA formulation. An LNP is primarily chemically and physically unstable [154]. Chemical instability is mainly shown by the degradation of lipids in the LNP that are susceptible to hydrolysis and oxidation, including oxidation due to hydrogen peroxide compounds (which can also lead to mRNA oxidation) [155] and hydrolysis of carboxylate bonds (DSPC and ionizable lipids). Physical instabilities are mainly LNP aggregation, fusion, and leakage [156].

The main obstacle to the distribution of the current mRNA COVID-19 vaccine is cold chain transport. Moderna’s new Spikevax™ COVID-19 vaccine can be stored for up to 6 months at the required storage temperature of −20 °C and for only 4 weeks at 2–8 °C. BioNTech/Pfizer’s Comirnaty™ vaccine can be stored at an ultra-low temperature of −70 °C for 6 months, at −20 °C for 2 weeks, and at 2 to 8 °C for only 5 days. It is unknown whether it is the mRNA or the LNP that causes the instability of the LNP@mRNA preparation. Compared with the LNP@mRNA system, the siRNA-LNP drug Onpattro™ has a significantly longer shelf life, suggesting that the current stability bottleneck is not the LNP but the mRNA. Linde Schoenmaker et al. [157] also pointed out that the self-stability of the mRNA molecule determines the storage conditions and shelf life of the current LNP@mRNA COVID-19 vaccines. To stabilize the mRNA, scientists employ RNase inhibitors to protect the mRNA from degradation [158]. Moderna also applies a hydroxyl radical scavenger (Tris-HCl buffer) that has an additional stabilizing effect on the nucleic acid macromolecules. Since mRNA is most stable in a weakly alkaline environment, Pfizer/BioNTech maintains the pH of its vaccines between 7 and 8. It is worth noting that the surface pH of ionizable cationic lipids may be higher than that of the surrounding aqueous medium [159]. In addition, purification of mRNA through fast protein liquid chromatography or high-performance liquid chromatography (HPLC) are also the main methods that are currently applied to improve mRNA stability by removing any remaining reaction byproducts, including the removal of immunogenic contaminants (dsRNA), and to avoid activation of RNA sensors, increase the translation level of encoded proteins, and reduce toxicity during repeated dosing. mRNA can be produced at a large scale and in accordance with good manufacturing practice processes [160]. Recently, Baierdorfer et al. developed a simple, highly scalable purification method for the removal of these transcriptional byproducts from samples [161]. They purified dsRNA via selective adsorption into cellulose in an ethanol-containing buffer and showed a comparable efficacy to HPLC methods, facilitating the large-scale production of mRNA therapeutics.

The CVnCoV candidate vaccine from CureVac is stable for storage at 5 °C for 3 months. CVnCoV utilizes Acuitas Therapeutics’ ionizable lipids and unmodified mRNA that encodes a full-length echinoderm protein with two proline substitutes. In the phase I clinical trial, volunteers produced neutralizing antibodies that were similar to those produced by patients recovering from COVID-19 and were well tolerated. Unfortunately, CVnCoV showed only a 47% efficacy in a phase III clinical trial that included 40,000 people. An interim analysis suggested that the lower efficacy of CVnCoV was attributed to the emerging SARS-CoV-2 variant rather than to the stability of the vaccine itself. CureVac and Abbott have developed the heat-resistant CV2CoV and ARCoV vaccines, respectively, to address this issue; both of these use unmodified mRNA and optimize the GC content for codons, which affects the secondary structure of the mRNA and may be responsible for the heat resistance of these two mRNA vaccines. However, CureVac’s previous-generation heat-resistant mRNA vaccine (CVnCoV) has failed, so the stability of the current heat-resistant mRNA vaccine remains to be proven.

Freeze-drying is also an important method to enhance the stability of LNP@mRNA vaccines. Freeze-drying is the process of removing water via sublimation at low temperatures under a vacuum. The lyophilized mRNA can be stored for long periods at 4 °C or at room temperature and has a relatively gentle drying method that improves the stability of the nanoparticles. In 2020, Moderna announced that its mRNA-1273 can be stored stably for up to 30 days at temperatures ranging from 2.22 °C to 7.78 °C (but once thawed, it cannot be refrigerated), while it can be stored for up to 6 months at −20 °C. The vaccine has received an emergency use authorization from the FDA and EMA due to an efficiency rate of 94.5%. Moderna recently completed a phase 1 clinical trial of the new-generation vaccine candidate mRNA-1283, which will be more optimized for storage temperature with the intention of improving the convenience of vaccine distribution and vaccination. It is possible that mRNA-1283, which encodes the stinger protein portion of the new SARS-CoV-2 coronavirus, is more stable, particularly the receptor binding domain and the N-terminal structural domain. Meanwhile, Moderna’s vaccine against cytomegalovirus infection (mRNA-1647) was offered in a frozen-liquid formulation in a phase I study but was changed to a lyophilized formulation in the Phase II study; the lyophilized formulation was reported to be stable for 18 months at refrigerated temperatures. When comparing of the expiration dates and storage conditions of the two currently marketed new crown mRNA vaccines, the role of the lyophilized formulations in enhancing the stability of the formulations can be considered more remarkable. BioNTech/Pfizer is also conducting a phase III clinical trial to test a lyophilized form of its BNT162b2 vaccine. Shifa also announced its SYS6006 vaccine, which is in phase II clinical research, with a new cationic lipid developed independently to optimize the LNP prescription and claiming long-term storage at 2 to 8 °C.

However, the drying process for LNP@mRNA is more complex due to mechanical forces generated during freezing and dehydration that can deform the carrier structure, which leads to carrier aggregation and mRNA breakage or leakage. In addition, it has been shown that even though LNP@mRNA maintains its integrity and encapsulation efficiency, the transfection efficiency in vivo after lyophilization is greatly reduced for some unknown reason, so the preparation should contain a lyophilization protectant to stabilize these particles during the freeze-drying process. Sugars such as sucrose or alginate are commonly used as lyophilization protectants to allow LNP@mRNA vaccines to be stored at a temperature closer to room temperature. Both the Moderna and Pfizer/BioNTech vaccines contain a similar high concentration of sucrose as a cryoprotectant. Moderna is frozen in two buffers (Tris and acetate), while the Pfizer/BioNTech vaccine uses only a phosphate buffer. It is well known that phosphate buffers are not suitable for freezing because they tend to precipitate and cause sudden changes in pH at the onset of ice crystallization. It is very challenging to figure out the lyophilization process for LNP@mRNA. At present, Pfizer has initiated a phase III study to compare the safety and efficacy differences between a lyophilized BNT162b2 formulation and its frozen-liquid BNT162b2 formulation. Researchers have lyophilized mRNA lipid-like nanoparticles (LLNs) with 5% sucrose or alginose and found that after lyophilization and re-solubilization, the LLNs did not lose activity in the in vitro assays, but the expression efficiency in vivo was significantly reduced [162]. It was speculated that it may be an alteration of the mRNA-LLN nanostructure that affected the binding of the nanoparticles to serum proteins. Leavit et al. found that when mRNA was lyophilized and re-solubilized and mixed with a blank LNP suspension, the mRNA was encapsulated by LNPs; the results showed that the mRNA was still active when administered [163], suggesting that neither lyophilization nor re-solubilization of the lipid nanoparticles interfered with the in vivo transfection efficiency of the mRNA. Recently, Reckitt Benckiser successfully developed the RH109 lyophilized mRNA vaccine [164], which has entered the clinical phase. They prepared three freeze-dried vaccines against the wild-type, Delta, and Omicron SARS-CoV-2 variants and demonstrated their ability to induce high levels of IgG titers and neutralization reactions as well as a long-term storage stability at 4 °C and 25 °C, which would solve the current storage and transport problems for mRNA vaccines, but the company did not publish the protective agents used in their freeze-dried vaccines or the details of its lyophilization process.

In addition, researchers have also attempted to use different delivery methods such as nebulization and oral administration to improve the stability and targeting of LNP-delivered mRNA. Zhang et al. indicated that nebulized LNP was stable for 14 days under storage conditions at 4 °C with no change in physical properties [165]. Melissa P. Lokugamage et al. described an iterative in vivo clustering-based screening method to identify LNP chemistry for improved lung delivery and constructed a novel LNP for efficient delivery of mRNA to the lungs via nebulization called Nebulized Lung Delivery 1 (NLD1), which could deliver mRNA that encoded broadly neutralizing antibodies more efficiently than the existing RNA-delivery systems and protected mice from lethal attack by influenza A (H1N1) subtype A viruses [166]. Notably, the team proposed three principles for the design of NLD1: (i) PEG lipids are essential to the formation of a stable LNP structure based on 7C1 (an oligomer–lipid conjugate); (ii) the combination of cationic co-lipids and a high molar percentage of PEG leads to increased mRNA delivery after aerosolization; and (iii) LNPs formulated with neutral phospholipids require less PEG than LNPs formulated with cationic auxiliary lipids. Although the vaccine was nebulized and delivered directly to the lungs by inhalation, which avoided the adverse effects of injections and required a much lower dose of vaccine, the nebulizer could cause the LNPs to produce aerosols, which led to LNP aggregation and a loss in the in vitro transfection activity [167]. Therefore, the choice of the route of administration is one of the challenges of nebulized LNP@mRNA. In addition, the stability of the structure of the nebulized LNPs during delivery and the different interactions with cells, proteins, and physical barriers in the trachea are not yet known. There are many patents on spray-dried LNP@mRNA, but polymers are required as stabilizers to ensure in vivo activity.

## 7. Optimization of LNP@mRNA Recipes for Clinical Therapeutic Requirements

### 7.1. Optimization of LNP@mRNA Recipes

Despite the great progress in the delivery of siRNA with LNPs, the optimal formulation composition and physicochemical properties of LNPs for mRNA delivery are not yet well defined due to the molecular differences between siRNA and mRNA. Therefore, optimizing the LNP@mRNA formulation is one of the most pressing issues that needs to be addressed.

To enhance the fundamental understanding of how the LNP structure affects mRNA delivery to target cells and tissues in vivo, Pedro P. Guimaraes et al. [168] designed an in vivo platform for accelerated mRNA delivery screening that consisted of an engineered LNP library that encapsulated functional, custom-designed barcoded mRNAs (barcode-mRNAs or b-mRNAs). These b-mRNAs were structurally and functionally similar to conventional mRNAs and contained barcodes that allowed them to be quantified by deep sequencing. They found that these different preparations could be combined together and injected intravenously into mice as a single pool, and they could be delivered to multiple organs (liver, spleen, brain, lung, heart, kidney, pancreas, and muscle) and simultaneously quantified using deep sequencing. In the context of liver and spleen delivery, LNPs with high b-mRNA delivery also produced high luciferase expression, suggesting that the platform could identify lead LNP candidates as well as the optimal formulation parameters for in vivo mRNA delivery. Interestingly, LNPs with the same formulation parameters that encapsulated different types of nucleic acid barcodes (b-mRNA vs. DNA barcodes) altered in vivo delivery, which implicated that the structure of the barcoded nucleic acids influenced the delivery of the LNPs in vivo. The platform enabled direct barcoding and subsequent quantification of the functional mRNA itself, which will allow for accelerated in vivo screening and the design of LNPs for mRNA therapeutic applications such as CRISPR/Cas9 gene editing, mRNA vaccinations, and other mRNA-based regenerative-medicine and protein-replacement therapies.

Kauffman et al. [103] developed a general strategy to optimize LNP formulations by varying the molar ratio of lipids and the nitrogen-to-phosphorus ratio by incorporating DOPE and increasing the weight ratio of ionizable lipids to mRNA via microfluidics and the DoE method. Relative to the original formulation, the optimized LNP formulation showed a 7-fold higher EPO expression in the liver compared to the original LNP formulation, which indicated that this optimization strategy approach can accelerate the screening in vivo and optimization of nanoparticle formulations with a large multi-dimensional design space. Interestingly, the optimized LNP did not improve siRNA delivery, which implied a difference in the design space of the optimized formulation parameters for siRNA and mRNA. In addition, Sato et al. [169] reported a two-step DoE approach to the preparation of liver-targeted, mRNA-loaded LNPs with pH-sensitive cationic lipids and an iLiNP device in which they optimized several formulation parameters including the nitrogen-to-phosphorus ratio, cationic lipid concentration, phospholipid concentration, and molar ratio of lipids and analyzed a variety of responses including the physicochemical properties, gene expression, and liver specificity. These results suggested that the LNP size and PEG/phospholipid ratio are key factors in promoting liver-specific gene expression and that this “multi-response DoE” approach can be applied to predict important parameters of LNP preparation and optimize the LNP@mRNA formulation more effectively.

Melissa P. Lokugamage et al. [166] designed an in vivo clustering-based iterative screening method to formulate six groups of LNP drugs with compound 7C1 (oligomer-lipid conjugate). Three principles were concluded based on the experiments: (1) PEGylated lipids are essential to the formation of stable 7C1-based LNP structures; (2) the combination of cationic auxiliary lipids and a high molar percentage of PEG leads to increased mRNA delivery after aerosolization; and (3) LNPs formulated with neutral phospholipids require less PEG than LNPs formulated with cationic auxiliary lipids. Using these three principles, the team constructed a novel LNP@mRNA system called NLD1 for further analysis and found that NLD1 was more stable and well tolerated after nebulization with more efficient transfection of the lungs and extensive expression of its delivered mRNA in the lungs.

### 7.2. Designing Novel PEGylated Lipids

Shielding particles with PEG is a recognized method to reduce the interaction of LNPs with serum proteins and complements [75]. The concentration of PEG in the system has the potential to reduce the immunogenicity of LNPs. Varun Kumar et al. [170] showed that increasing the molar concentration of PEG lipids in LNPs from 1.5 to 10% reduced cytokine production, complement activation, and macrophage recognition of the LNPs, but as expected, it also inhibited the interaction of LNPs with ApoE, thus affecting their efficiency in gene transfection. The study showed that it is important to determine the PEG molar concentration that allows for the maximum protection of the LNPs from the immune system without sacrificing their activity. A highlight of the study was the introduction of PEG lipids only after LNP formation with the aim of ensuring that the screened LNPs were similar in size while obtaining the desired PEG surface density.

However, PEGylated lipids tend to cause ABC effects. The replacement of LNPs with PEGylated liposomes or polymeric nanoparticles [171], thereby reducing the ABC effect of LNPs, was also studied in the past, but the development of this research was hampered by the development of resistance to PEG lipids in humans, leading to the potency of LNP@mRNA or resulting in increased allergic reactions [153]. Recent studies showed that stable LNP delivery systems can be achieved with alternative materials to PEGylated lipids. For example, LNPs prepared using 3% Tween 20 in place of PEGylated lipids were able to specifically deliver pDNA to the lymph nodes of mice, which was superior to standard preparations with DSPE-PEG2000, thereby enhancing transfection of this target organ. In addition, Tween 80 with a long lipid tail could be applied to form a stable LNP for spleen targeting, but the transfection efficiency was relatively low [172]. Polysarcosine (pSar) is a polymer of the endogenous substance sarcosine. Sara S. Nogueira et al. [152] employed pSar instead of PEG to achieve reduced LNP aggregation and RES clearance in vivo, which improved the RNA transfection efficiency, reduced pro-inflammatory factor release, and decreased complement activation. The application of pSar lipids with different polymer chain lengths and molar fractions enabled control of the physicochemical properties of the LNPs such as the particle size, morphology, and internal structure. In combination with suitable ionizable lipids for the assembly of LNPs, it showed a high RNA transfection efficacy and an improved safety profile after intravenous administration. Notably, in comparison with the PEGylated lipid-based system, higher protein secretion and reduced immunostimulatory responses were observed. In summary, pSar-based LNPs are able to deliver mRNA safely and effectively, thus providing a good basis for the development of novel PEG-free mRNA therapies. To date, other companies including BioNTech and Arcturus Therapeutics have begun attempts to remove PEG from LNPs.

## 8. The Brief Landscape of LNP@mRNA in Clinical Gene Therapy

Gene therapy has been a hot clinical therapeutic approach in recent years. Nucleic acids based on lipid nanoparticles available for gene therapy include antisense oligonucleotides (ASO), short interfering RNA (siRNA), microRNA (miRNA), and mRNA, which have various mechanisms of action. ASO functions through RNAse-H-mediated action to hybridize and cleave the target mRNA. Differently, siRNA and miRNA inhibit mRNA translation through the RISC-mediated RNAi pathway, thereby preventing protein synthesis. In contrast, mRNAs are bound with ribosomes via cap and poly(A) binding proteins and then are translated into therapeutic proteins, which is the basis for the mechanism of mRNA in LNP@mRNA vaccines.

In view of the excellent safety and efficacy of mRNA vaccines with LNPs as a delivery vehicle for the prevention of COVID-19, the LNP@mRNA delivery system is beginning to be widely used for the treatment of various diseases including cardiovascular diseases, cancer, and infectious diseases [173]. On the clinicaltrials.gov website, the current mRNA clinical trials, after removing the COVID-19 mRNA trials, total 38 new therapeutic trials. Of these, cancer accounts for 50%, infectious diseases for 29%, and genetic diseases for 18.4%, with LNPs being the most popular delivery vector. Table 5 shows the current clinical trials of LNP@mRNA.

### 8.1. LNP@mRNA for Infectious Disease Treatment

The mRNA vaccines can be applied in both prophylactic and therapeutic vaccinations. In comparison with protein or DNA vaccines, mRNA can serve as a prophylactic against diseases for which conventional vaccines have not shown sufficient efficacy due to the properties of immune activation and the amount of antigen that can be delivered. With short production times, mRNA vaccines can also be used to provide a rapid response to emerging threats or seasonal strains of pathogens. The respiratory syncytial virus fusion glycoprotein (RSV-F) is a conserved target for neutralizing antibodies and the most promising antigen for the development of RSV vaccines. Recently, Espeseth et al. [174] formed an LNP@mRNA vaccine by encapsulating chemically modified mRNA with LNPs that has good immunogenicity and protection against RSV. As compared with protein-based vaccines, the mRNA/LNP vaccine triggered significant CD4^+^ and CD8^+^ T-cell responses in mice and provoked a strong cellular immune response against RSVF. This LNP@mRNA vaccine that expresses the RSVF protein has the potential for safe and effective application in the prevention of RSV disease. In January 2020, CureVac reported interim safety and immunogenicity data from a phase I study of CV7202, a novel prophylactic LNP@mRNA vaccine against rabies (using LNPs provided by Acuitas). Two 1 or 2 μg doses of CV7202 were well tolerated and elicited rabies-neutralizing antibody responses in all recipients, which met the WHO’s protection criteria [175]. Similarly, Moderna is highly active in the field of LNP@mRNA vaccines for many infectious diseases. Thus, in early 2020, Moderna, BioNTech/Pfizer, and CureVac were poised to rapidly initiate a clinical mRNA vaccine program to address SARS-CoV-2. Currently, vaccine candidates for several viruses including the Nipah virus, human immunodeficiency virus (HIV), Epstein–Barr virus, and seasonal influenza are currently in clinical development.

LNP@mRNA vaccines can also be adopted in the veterinary field to prevent infectious diseases in animals. Saxena et al. [176] applied a self-amplified mRNA vaccine that encoded the rabies virus glycoprotein to induce an immune response for the prevention of canine rabies. Recently, vanBlargan et al. [177] developed an LNP to encapsulate a modified mRNA vaccine (with prM and E genes that encoded the deer Powassan virus) to form an LNP-mRNA vaccine platform and demonstrated its immunogenicity and efficacy after the vaccination of mice. The platform has utility in the development of vaccines against various flaviviruses (from mosquito- and tick-borne vectors) and was further optimized to make the immune response more adaptive to achieve broad protection against multiple viruses.

### 8.2. LNP@mRNA for Cardiovascular Disease Treatment

Heart failure is caused by activated cardiac fibroblasts to some extent; these respond to heart injury and inflammation by chronically overproducing fibrous material that hardens the heart muscle and impairs heart function (a process known as cardiac fibrosis). Therefore, scientists envisioned whether a (CAR)-T cell could be designed to attack activated cardiac fibroblasts. Most of the (CAR)-T therapies currently on the market or under development involve isolating T cells from patients; the T cells are then modified in vitro to express (CAR)-T cells that recognize cancer cell surface antigens such as CD19 and BCMA followed by amplifying the (CAR)-T cells and finally infusing them back into the patient to play a role in destroying the cancer cells.

However, there are unique challenges in adopting this traditional (CAR)-T cell therapy to treat human heart failure or other fibrotic diseases compared with fighting cancer because fibroblasts themselves have normal and important functions in the body, especially in wound healing. (CAR)-T cells can survive for months or even years after infusion into a patient, so they may continue to suppress the number of fibroblasts and impair wound healing. In cancer therapy, the long-term presence of (CAR)-T cells is an advantage because they provide long-lasting efficacy. However, if a patient is injured after receiving conventional (CAR)-T therapy, the long-term presence of (CAR)-T cells may interfere with wound healing and potentially bring safety risks. To address this issue, researchers [125] published a breakthrough in (CAR)-T for heart injury that was on the cover of *Science*. They designed an mRNA that encoded the CAR that binds to fibroblast-activation proteins expressed on the surface of fibroblasts, after which they encapsulated the mRNA in an LNP, which in turn had an antibody coupled to its surface that could target CD5T cells. Studies have shown that therapeutic (CAR)-T cells can be generated in vivo intact via injection of CD5-targeted LNPs containing mRNA. Analysis of a mouse model of heart failure showed that this innovative therapeutic approach was successful in reducing fibrosis as well as restoring cardiac function. This achievement represents not only an important breakthrough in the expansion of (CAR)-T indications, but also a milestone advancement in the field of in vivo (CAR)-T therapy. Researchers believed that the application of modified mRNA to generate (CAR)-T cells directly in vivo may have a wide range of therapeutic applications.

### 8.3. LNP@mRNA for Liver Disease Treatment

Rizvi et al. [178] developed a nucleoside-modified mRNA (LNP@mRNA) wrapped by LNP, which transiently and stably expressed hepatocyte growth factor (HGF) and epidermal growth factor (EGF) in mouse hepatocytes. Meanwhile, they confirmed the specific hepatic targeting of LNP@mRNA by intravenous injection of LNP@mRNA encoding firefly luciferase, which resulted in the stable expression of protein levels for about 3 d. In addition, HGF LNP@mRNA can effectively induce hepatocyte proliferation. In a mouse model of chronic liver injury with non-alcoholic fatty liver disease and a mouse model of acute liver injury induced by acetaminophen, the injection of HGF and EGF LNP@mRNA significantly reversed steatosis, rapidly activated regenerative pathways in hepatocytes, and accelerated the recovery of liver function after injury. Therefore, LNP@mRNA is an ideal therapeutic strategy to activate the regenerative pathway of hepatocytes for the treatment of multiple causes of liver injury in a short period of time.

### 8.4. LNP@mRNA Research for the Cancer Treatment

Due to its property of rapid synthesis, mRNA has become a very suitable choice for making personalized vaccines in the context of precision medicine, especially for diseases that are extremely heterogeneous, such as cancer. With the ability to induce an immune-cell-mediated response that generates a strong CD8^+^ T-cell response to clear or reduce tumor cells, the mRNA vaccines mediated by LNPs are now considered to be highly promising cancer therapies. Currently, there is a great deal of research on the development of cancer mRNA vaccines [179]. Cancer mRNA vaccines can express tumor-associated antigens that stimulate cell-mediated immune responses to remove or inhibit cancer cells. Consequently, most cancer mRNA vaccines are being increasingly employed as therapeutic agents than as prophylactic agents [180]. Recent studies also developed mRNA vaccines with anti-cancer effects that induced strong and potent T-cell and humoral immune responses. Lee et al. [181] selected tripalmitoyl-*S*-glycero-cysteine linked to a pentapeptide (PAM3CSK4; Pam3) as an adjuvant to encode ovalbumin mRNA bound in LNPs to develop a Pam3-doped mRNA/Pam3-LNP vaccine. Since Pam3 can be recognized by Toll-like receptors (TLR) 2 and 1 on tumor cell membranes, it can greatly improve the efficiency of mRNA/Pam3-LNP vaccine uptake. After entering cells through endocytosis, the mRNA/Pam3-LNP vaccine dissociates under acidic conditions to release the mRNA, which can be recognized by TLR7/8 on the endosomal membrane. Based on the synergistic effect of Pam3 and mRNA triggering different TLR subclasses, the Pam-LNPs exhibited strong immune stimulation in cells and produced a large number of antigen-specific CD8^+^ T cells, which greatly improved the efficacy of the mRNA vaccine in tumor prevention.

In addition, studies have shown that dysfunction caused by mutations in the p53 gene led to immunosuppression and immune escape. Therefore, restoring p53 function provides an opportunity to reverse immunosuppression in the tumor microenvironment and improve the efficacy of immune-checkpoint therapies. Based on this, researchers [182] developed lipid nanoparticles consisting of G0-C14, PLGA, and lipid PEG and modified them with CTCE-9908, a targeting peptide specific to CXCR4 (which is highly expressed in hepatocellular carcinoma cells) to achieve specific targeting of hepatocellular carcinoma cells. The application of this novel lipid nanoparticle delivery vehicle for the specific delivery of p53mRNA to hepatocellular carcinoma cells in combination with an anti-PD-1 monoclonal antibody could effectively induce global reprogramming of the tumor microenvironment, which resulted in better anti-tumor effects. The findings suggested that the combination of the p53 mRNA nanodrug and immune-checkpoint blockade therapy can reverse immunosuppression in hepatocellular carcinoma, which is expected to be a breakthrough cancer treatment.

Antibodies against T-cell costimulatory receptors have been developed to activate T-cell immunity in cancer immunotherapy. However, tumor-infiltrating immune cells often lack the expression of co-stimulatory molecules, thus hindering antibody-mediated immunotherapy. Recently, researchers [183] designed and synthesized phospholipid and glycolipid derivatives (PLs and GLs) and constructed a library of phospholipid and glycolipid mimetic materials based on phospholipids and glycolipids, which are natural components of cell membranes. These compounds consisted of a mimetic head (phosphate head or glycosyl head), an ionizable amino core, and multiple hydrophobic tails. A total of 18 phospholipid derivatives and 16 glycolipid derivatives were synthesized; phospholipid derivative 1 (PL-1) showed the best performance. PL1 nanoparticles could deliver co-stimulatory receptor mRNA not only to T-cell lines in vitro, but also to T cells within tumors in vivo, thereby providing useful delivery materials for regulating T-cell functions. Agonistic antibodies against co-stimulatory receptors that enhance anti-tumor T-cell immunity have been used in cancer therapy. Subsequently, to enhance tumor suppression, the authors replaced wild-type OX40 mRNA with pseudouracil (ψ)-modified mRNA and increased the amount of mRNA delivered per injection from 8 µg to 40 µg. With systemic administration, the experimental results in mice with lung metastases demonstrated that these improvements significantly enhanced the effect of this immunotherapy.

Since the FDA approval of (CAR)-T in 2017, an increasing number of leukemia patients and lymphoma patients have experienced complete remission. (CAR)-T therapy relies on the ex vivo manipulation of a patient’s T cells to produce effective cancer-targeted therapies in patients with acute lymphoblastic leukemia and large B-cell lymphoma. However, current (CAR)-T cell engineering approaches use viral delivery vectors, which induce permanent CAR expression and can lead to serious adverse effects. Michael Mitchell et al. discovered that a new engineering technique that is less toxic to T cells could alter the way they recognize cancer through a different mechanism [184]. This new engineering technique utilized LNPs to deliver mRNA across the cell membrane of T cells rather than using a genetically modified virus to rewrite the DNA of T cells. This approach is certainly preferable to the current standard method of enabling mRNA across the cell membrane (i.e., electroporation), which may be too toxic to obtain the required number of T cells from the patient. Here, they synthesized a library of 24 ionizable lipids formulated as LNPs and screened for luciferase mRNA delivery to Jurkat cells, which revealed seven agents capable of enhancing mRNA delivery. The best performing LNP preparation (C14-4) was selected for delivery of CAR mRNA to the primary T cells. The platform induced CAR expression at levels equivalent to electroporation with a significantly reduced cytotoxicity. (CAR)-T cells engineered via C14-4LNP treatment were then compared with electroporated (CAR)-T cells in a co-culture assay with Nalm-6 acute lymphoblastic leukemia cells; both CAR-T cell engineering approaches elicited potent cancer killing activity. These results demonstrated the ability of LNPs to deliver mRNA to primary human T cells to induce functional protein expression and showed the potential of LNP to enhance mRNA-based (CAR)-T cell-engineering approaches.

Currently, there is also little progress in (CAR)-T treatment for solid tumors. The main reason is that there are many difficulties in cellular therapy for solid tumors, such as the large heterogeneity of different types of solid tumors, the lack of unique tumor-associated antigens as (CAR)-T targets, the inability of T cells to effectively target tumor sites, the limited persistence of (CAR)-T cells, and the complex microenvironment within the tumor (which has a suppressive effect on immunity). Apart from the huge bottleneck in efficacy, severe cytokine release syndrome and neurotoxicity (immune effector cell-associated neurotoxicity syndrome, cap structure (cap structure ICANs)) are also a major issue that plagues the use of (CAR)-T therapy for solid tumor treatment. In addition, (CAR)-T in solid tumors faces other safety risks such as macrophage activation syndrome, uveitis, etc. [185]. To address these drawbacks, mRNA technology can utilize the target site or antigen-encoding mRNA to make the encoded antigen available for cellular uptake and expression through a specific delivery system, thereby eliciting both humoral and cell-mediated immune responses. The utilization of nucleoside-modified mRNAs can carry temporary copies of genetic information that can be delivered via LNPs to produce a more controlled and shorter-lasting transient (CAR)-T therapy. The combination of LNP-delivered (CAR)-T and mRNA technology overlays will help drive (CAR)-T therapies to address the challenges of complex processes, long lead times, and high prices. On the other hand, the in vivo creation of functional T cells is expected to greatly extend the platform’s application prospects. BioNTech has pioneered the development of (CAR)-T mRNA vaccines with BNT211, a next-generation (CAR)-T therapy that targets solid tumors and combines CAR-T therapy targeting the claudin 6 (CLDN6) antigen with CARVac, an mRNA vaccine that expresses the CLDN6 antigen [186]. The initial data in patients with advanced solid tumors showed good safety and treatment outcomes with a disease control rate of 86% and an overall remission rate of 43% [187].

### 8.5. LNP@mRNA for Rare Disease Treatment

In patients with sepsis, macrophages and other immune cells are present in lower than normal numbers and do not function properly. Researchers [80] developed nanoparticles based on vitamins that were particularly adept at delivering messenger RNAs and constructed messenger RNAs that encoded anti-microbial peptides and signaling proteins, which enabled the specific accumulation of antimicrobial peptides in lysosomes inside macrophages, which are key to bacterial killing activity. Then, the nanoparticles loaded with mRNA were transported to macrophages generated from donor monocytes, which were then allowed to “make” therapeutic drugs from them. Since macrophages naturally have antimicrobial activity, if other antimicrobial peptides were added to the cells, they could further enhance the antimicrobial activity and help the entire macrophage to clear bacteria. The mouse model of sepsis in this study was infected with multi-drug-resistant Staphylococcus aureus and *Escherichia coli* and their immune systems were suppressed. Each treatment contained approximately 4 million engineered macrophages, and the control group consisted of normal macrophages and a placebo. Compared with the control group, the treatment significantly reduced the number of bacteria in the blood after 24 h; for those remaining in the blood, a second treatment cleared them.

Gene therapy has been extensively studied for the treatment of retinal degenerative diseases caused by advanced glaucoma, atrophic macular degeneration, advanced diabetic retinopathy, and hereditary retinal degeneration [188]. With the approval of an AAV-based gene therapy by the FDA for the treatment of congenital hazel disease, Vortigern Neparvovec (Luxturna) has made AAVs the standard for retinal delivery [189]. However, AAVs have many safety issues as viral vectors [190], and their inability to load large nucleic acids (>5 kb) limits their wider application. There is a need to develop delivery systems that overcome these limitations and expand the utility of gene therapy for retinal degeneration. Siddharth Patel et al. [191] tested 11 different LNP variants to determine their ability to deliver mRNA to the posterior part of the eye, among which LNPs containing ionizable lipids with low p*K*a values and unsaturated hydrocarbon chains showed the highest amount of reporter gene transfection in the retina. Gene delivery is cell-specific and is mostly expressed in the retinal pigment epithelium (RPE) with limited expression in Müller glial cells. The mRNA delivered by LNPs can be used to treat monogenic retinal degenerative diseases of the RPE, and the transient property makes them suitable for applications that target retinal reprogramming or genome editing. Overall, the delivery of mRNA to various cell types within the retina with LNPs could provide a transformative new approach to preventing blindness.

The application of mRNA drugs for cancer treatment with the ability to simultaneously detect and image tumors would be even more beneficial to tumor treatment. Therefore, the development of therapeutically integrated LNP@mRNA that can efficiently produce functional proteins synchronized with tumor imaging is an exciting and unresolved challenge. Daniel J. Siegwart et al. [192] prepared a dendrimer-based LNP (DLNP) system for therapeutic and diagnostic purposes based on this that contained the PEGylated fluorescent dye BODIPY for the delivery of mRNA and near-infrared imaging in vitro and in vivo. The DLNPs successfully mediated the expression of the tumor mRNA and simultaneously illuminated tumors in pH-sensitive NIR imaging. This therapeutic lipid nanoparticle that combines both mRNA delivery and NIR imaging is expected to be a viable method for the simultaneous detection and treatment of cancer in the future.

## 9. Conclusions and Perspective

To exert therapeutic effects more effectively in vivo, it is necessary to select an appropriate vector to deliver mRNA. There are several requirements for mRNA vaccine-delivery systems: a high encapsulation rate and protection of the mRNA from degradation by hydrolytic enzymes; the capability to deliver the vaccine to target cells and release mRNA to the cytoplasm for translation efficiently; and a low toxicity without causing severe inflammatory reactions or toxic side effects in the organism. Non-viral-vector delivery systems, as represented by LNPs, can effectively load mRNA and transfect cells to express specific antibodies and activate the body’s immune response, thereby exhibiting good clinical results in response to COVID-19 infection. Meanwhile, the clinical results after utilization through emergency authorizations also confirmed the high safety of the mRNA vaccines delivered using LNPs as a carrier. Currently, on the clinicaltrials.gov website, most RNA clinical trials focus on mRNA (53.2%), and LNPs are preferred as the delivery vehicle (51.4%). Therefore, delivery of mRNA by LNPs is a hot research topic at present. An analysis of disease types showed that infectious diseases (43.2%), genetic diseases (25.7%), and cancer (14.9%) are the three hottest research directions at present, and LNP@mRNA also has become a novel agent for the treatment and prevention of infectious diseases and cancer at present. With the continuous improvements in mRNA technology and LNP delivery technology, this novel technology will help solve the problems that traditional small-molecule and antibody therapies cannot address by providing a more effective and durable therapeutic technology option for infectious diseases, oncology, cardiovascular diseases, endocrine system diseases, and other diseases.

However, there are still many problems in the application of LNP@mRNA. LNP@mRNA vaccines are extremely unstable and degrade quickly at room temperature, therefore requiring ultra-low temperature storage and harsh storage conditions. Therefore, it is important to solve the problems of long-term storage and long-distance transportation of LNP@mRNA. In addition, while ionizable lipids within the LNPs are protonated at a low pH to promote electrostatic interactions with the anionic endosomal membrane, thereby triggering the release of nucleic acids into the cytoplasmic matrix [193], studies have shown that most LNPs are trapped within degradative endocytic compartments or efflux from cells and only <2% reach the cytoplasmic matrix [194]. The endosomal escape process remains one of the least known and least cooperative barriers to successful gene delivery [195]. In addition, the current development of LNP@mRNA drugs is still immature and both in terms of sequence design and nucleic acid modification technology, the delivery system platform is in the early stages. There are many outstanding questions about LNPs: How do the properties of lipid membranes affect the transfection efficiency? What is the localization of each lipid fraction in LNPs? Are lipids separated during long-term storage? How do nucleic acids interact with ionizable lipids? How does the internal structure of LNPs prevent nucleic acid degradation? What factors affect the amount of RNA in each LNP? Therefore, there is still a long way to go in the clinical application of LNP@mRNA.

Encouragingly, the success of the LNP@mRNA vaccines in treatment of COVID-19 has shined a light on their effectiveness. Further research on the surface characteristics of LNPs, the form of administration, the mode of uptake, endosomal escape, carrier release, the dose, RES clearance, and safety will further motivate the development of LNP@mRNA in the field of gene therapy [196]. It is believed that LNP@mRNA formulations will gain a wider scope of applications and become a new breakthrough point in biopharmaceuticals.

## Figures and Tables

**Figure 1 pharmaceutics-14-02682-f001:**
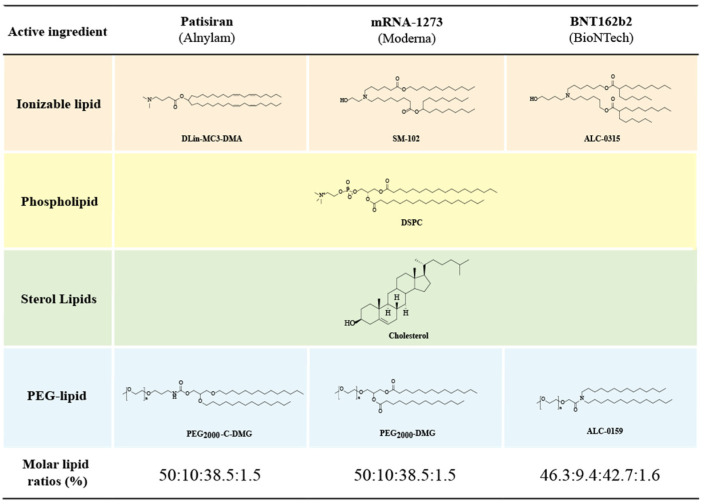
Structure and molar ratio of the component lipids in the three clinical applications of LNPs. Reprinted with permission from Ref. [61]. Copyright © 2021 The Japanese Society for the Study of Xenobiotics. Published by Elsevier Ltd.

**Figure 2 pharmaceutics-14-02682-f002:**
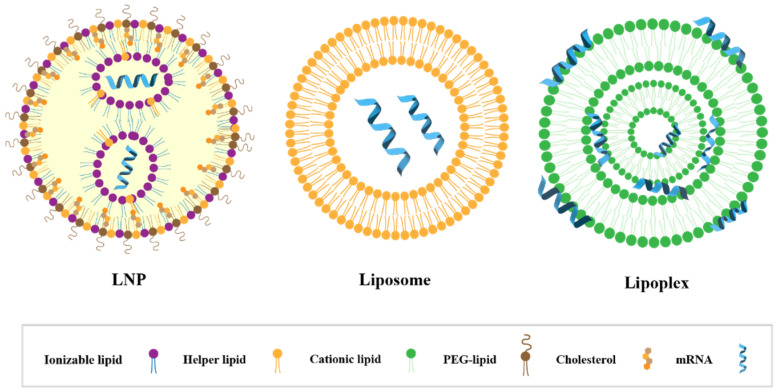
The schematic illustrations of the LNP, liposome, and lipoplex for mRNA delivery.

**Figure 3 pharmaceutics-14-02682-f003:**
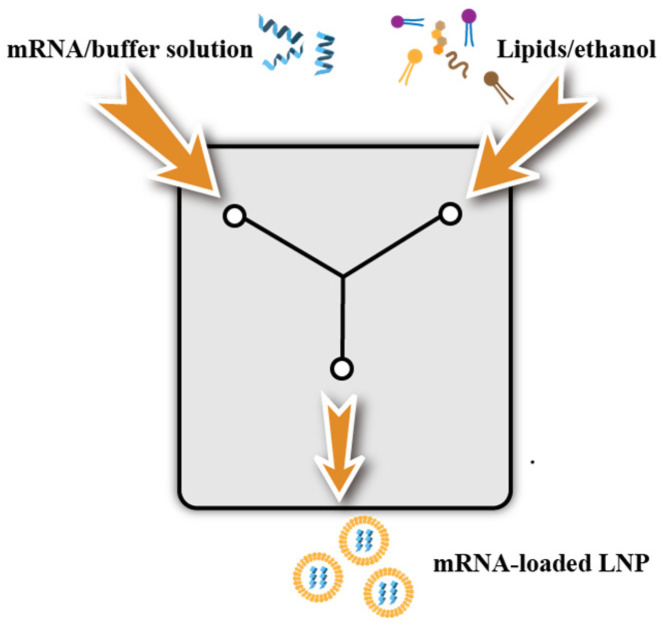
Diagram of LNP production with microfluidic device.

**Figure 4 pharmaceutics-14-02682-f004:**
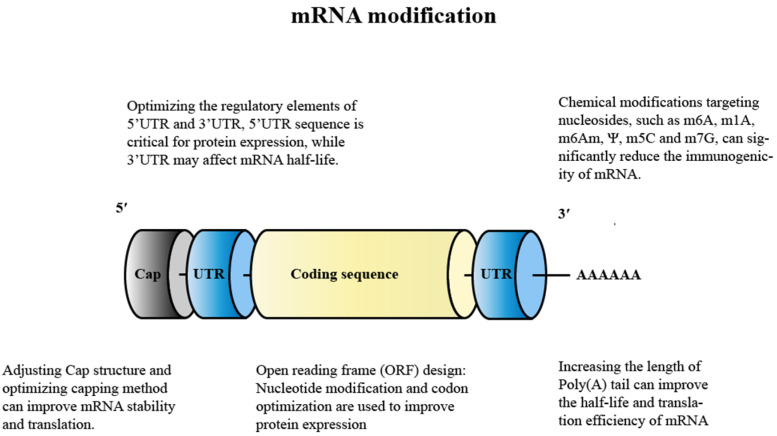
Principles and methods of modifying mRNA. Reprinted with permission from Ref. [73]. Copyright © 2021 Elsevier Ltd.

**Figure 5 pharmaceutics-14-02682-f005:**
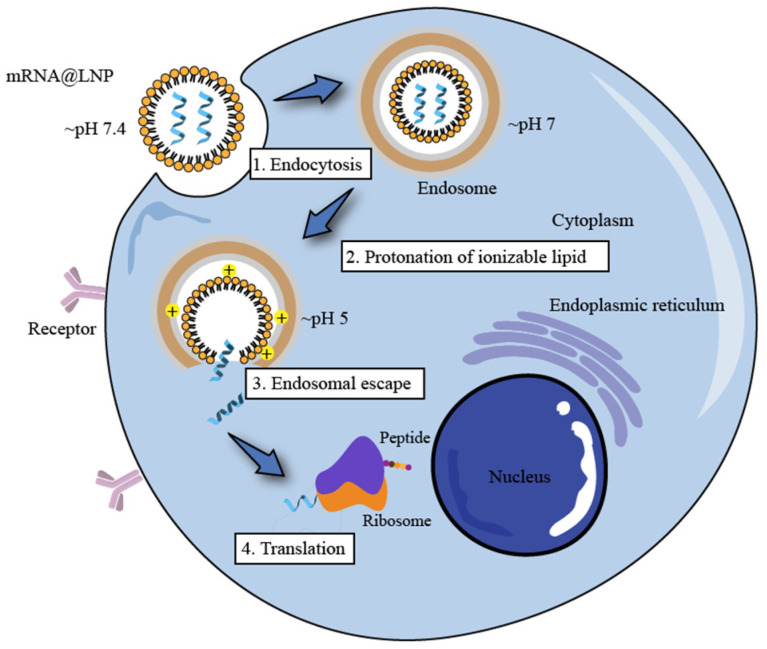
Delivery mechanism of LNP@mRNA.

**Figure 6 pharmaceutics-14-02682-f006:**
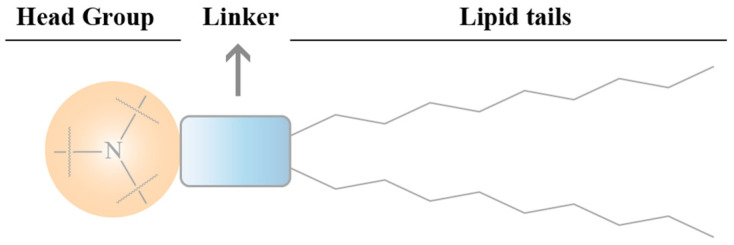
Schematic illustration of ionizable lipid structure.

**Figure 7 pharmaceutics-14-02682-f007:**
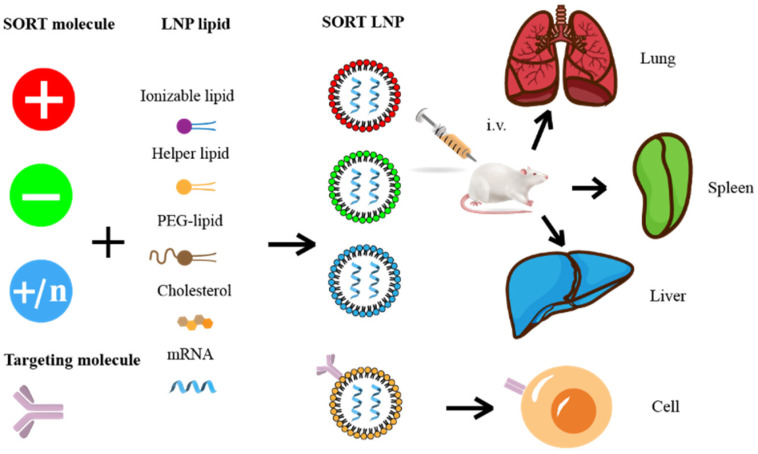
Delivery strategies for various ionized lipids and ligands.

**Table 1 pharmaceutics-14-02682-t001:** The mRNA vaccines for COVID-19 that have been approved or in clinical trials.

Technology Path	Vaccine Name	Companies	First Posted	Clinical Trial Number	Phase
mRNA vaccines	Comirnaty BNT162B2	Pfizer (New York, NY, USA)/BioNTech SE (Mainz, Germany)	Approved. Emergency use authorization in several countries around the world		
	Spikevax mRNA-1273	ModernaTX, Inc. (Cambridge, MA, USA)			
	mRNA-1273.351	ModernaTX, Inc. (Cambridge, MA, USA)	28 May 2020	NCT04405076	Phase II
	SYS6006	CSPC ZhongQi Pharmaceutical Technology Co., Ltd. (Shijiazhuang, China)	30 June 2022	NCT05439824	Phase II
		CanSino Biologics Inc. (Tianjin, China)	13 May 2022	NCT05373472	Phase II
	ABO1009-DP	Suzhou Abogen Biosciences Co., Ltd. (Suzhou, China)	28 June 2022	NCT05434585	Phase I
	ABO-CoV.617.2				
	LVRNA009	AIM Vaccine Co., Ltd. (Beijing, China)	6 May 2022	NCT05364047	Phase I
	RQ3013	Walvax Biotechnology Co., Ltd. (Kunming, China)	31 May 2022	NCT05396573	Phase I
	PTX-COVID19-B	Providence Therapeutics Holdings Inc. (Toronto, ON, Canada)	21 February 2021	NCT04765436	Phase I
	CV0501	GlaxoSmithKline (Brentford, England)	28 July 2022	NCT05477186	Phase I
Lyophilized mRNA vaccine	RH109	Wuhan Recogen Biotechnology Co., Ltd. (Wuhan, China)	9 May 2022	NCT05366296	Phase I

**Table 2 pharmaceutics-14-02682-t002:** Advantages and disadvantages of viral and non-viral vectors.

	Viral Vectors	Non-Viral Vectors
Advantages	High transfection efficiency	Low toxicity, low immune response, low chance of exogenous gene integration, no size limitation of gene insert, easy to use, easy to prepare, easy to store and test, high safety, high potential, low cost, simple preparation and modifiability
Disadvantages	Potentially carcinogenic, autoimmunogenicity, cytopathic changes, small genetic capacity, toxic side effects, high preparation costs	Low transfection efficiency

**Table 4 pharmaceutics-14-02682-t004:** Characteristics of the current laboratory methods for LNP preparation.

Name	Cost	Scalability	Encapsulation Efficiency	Reproducibility	Polydispersity Index
Ethanol dilution method	Low	Moderate	Moderate	Moderate	High
Manual mixing method	Low	Low	Low	Low	High
T-mix method	Low	High	High	High	Moderate
Microfluidics	High	High	High	High	Low

**Table 5 pharmaceutics-14-02682-t005:** Current clinical trials of LNP@mRNA.

Institution	Name	Indication	mRNA Encoding	Route of Administration	Delivery Vector	Clinical Trial Number (Phase)
ModernaTX, Inc.	mRNA-1647	Cytomegalovirus (CMV) Infection	Human cytomegalovirus envelope glycoprotein H	i.m.	LNP V1GL	NCT03382405 (I) NCT04232280 (II)
	mRNA-1443				LNP	NCT03382405 (I)
	mRNA-1653	Human metapneumovirus (hMPV) and human parainfluenza 3 virus (PIV3)	hMPV and PIV3 membrane fusion protein	i.m.	LNP	NCT03392389 (I) NCT04144348 (I)
	VAL-506440; mRNA-1440	Influenza A virus (H10N8)	Influenza hemagglutinin H10N8	i.m.	LNP	NCT03076385 (I)
	VAL-339851; mRNA-1851	Influenza A virus (H7N9)	Influenza hemagglutinin H7N9	i.m.	LNP	NCT03345043 (I)
	mRNA-1345	Respiratory syncytial virus (RSV)	RSV pre-infused F protein	i.m.	LNP	NCT04528719 (I)
	VAL-181388; mRNA-1388;	Chikungunya virus (CHIKV)	Anti-CHKV monoclonal antibody	i.m.	LNP	NCT03325075 (I)
	mRNA-1944	Prevention of CHIKV infection	CHIKV-specific monoclonal neutralizing antibody (CHKV-24)	i.m.	LNP	NCT03829384 (I)
	mRNA-1325;	Zika virus	PrM and E	i.m.	LNP	NCT03014089 (I)
	mRNA-1893			i.m.	LNP V1GL	NCT04064905 (I)
	mRNA-2752	Solid tumor malignancies or lymphoma/ovarian cancer	Human OX40L, IL-23, and IL-36γ	Intratumoral injection	LNP	NCT03739931 (1)
	mRNA-2416		Human OX40L	Intratumoral injection	LNP	NCT03323398 (I/II)
	mRNA-4157	Melanoma	Personalized	i.m.	LNP	NCT03897881 (II)
		Solid tumors	Personalized	i.m.	LNP	NCT03313778 (I)
BioNTech SE	Lipo-MERIT	Melanoma	Four selected malignant melanoma-associated antigens: New York-ESO 1 (NY-ESO-1), tyrosinase, melanoma-associated antigen A3 (MAGE-A3), and trans-membrane phosphatase with tensin homology (TPTE)	i.m.	Lipoplex	NCT02410733 (I)
CureVac	CV7202	Rabies	Rabies virus glycoprotein (RABV-G)	i.m.	LNP	NCT03713086 (I)
GlaxoSmithKline	RG-SAM (CNE) vaccine (GSK3903133A)	Viral diseases	Rabies glycoprotein G (RG)	i.m.	CNE	NCT04062669 (I)

## Data Availability

Not applicable.

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
