# Peer review of "Recent Advances in Lipid Nanoparticles for Delivery of mRNA"

_pharmaceutics, 2022, doi:10.3390/pharmaceutics14122682_

Round 1

Reviewer 1 Report

The manuscript entitled ‘Recent Advances in Lipids Nanoparticles for Delivery of mRNA’ describes the importance of Lipids based Nanoparticles for Delivery of mRNA for development of vaccines and how these LNP may play an important role in the treatment of diseases. Below are few issues which authors should address:

1.      In table 1 authors should only focus on mRNA based vaccines.

2.      Authors should propose the mechanism of action of siRNA based Lipid Nanoparticles in different types of diseases.

3.      Authors should also described the mechanism of mRNA in vaccine development this will be clear to readers.

Manuscript is well written and are within scope of journal. This manuscript should be accepted after minor revision

Reviewer 2 Report

I recommend the authors see the attached PDF document for more clarity regarding subscripts, configurational descriptors, and alignments. 

Round 2

Reviewer 2 Report

1. In figure 2: There are spelling mistakes. 

lipoplexe -> lipoplex

polyplexe -> polyplex

2. Polyplex is not a type of LNP.

Polyplex is belong to the family of polymeric nanoparticles. 

3. Figure1 and Figure 6 contain the same chemical structures for ALC0135 and SM-102.

4. Fig 5: Released from endosome --> Endosomal escape
